# Bubble reachers and uncivil discourse in polarized online public sphere

**Jordan K. Kobellarz[1], Milos Brocic[2¤], Daniel Silver[2]\*, Thiago H. Silva**[1]

**1** Informatics, Universidade Tecnológica Federal do Paraná, Curitiba, Brazil, **2** Sociology, University of Toronto, Toronto, Canada

¤ Current address: Sociology, McGill University, Montreal, Canada
\* dan.silver@utoronto.ca

**Data Availability Statement:** The data is publicly available at https://doi.org/10.5281/zenodo.1044022. The code to replicate all analyses is available at https://github.com/jordankobellarz/bubble-reachers-incivility.

## Abstract

Early optimism saw possibilities for social media to renew democratic discourse, marked by hopes for individuals from diverse backgrounds to find opportunities to learn from and interact with others different from themselves. This optimism quickly waned as social media seemed to breed ideological homophily marked by "filter bubbles" or "echo chambers." A typical response to the sense of fragmentation has been to encourage exposure to more cross-partisan sources of information. But do outlets that reach across partisan lines in fact generate more civil discourse? And does the civility of discourse hosted by such outlets vary depending on the political context in which they operate? To answer these questions, we identified bubble reachers, users who distribute content that reaches other users with diverse political opinions in recent presidential elections in Brazil, where populism has deep roots in the political culture, and Canada, where the political culture is comparatively moderate. Given that background, this research studies unexplored properties of content shared by bubble reachers, specifically the quality of conversations and comments it generates. We examine how ideologically neutral bubble reachers differ from ideologically partisan accounts in the level of uncivil discourse they provoke, and explore how this varies in the context of the two countries considered. Our results suggest that while ideologically neutral bubble reachers support less uncivil discourse in Canada, the opposite relationship holds in Brazil. Even non-political content by ideologically neutral bubble reachers elicits a considerable amount of uncivil discourse in Brazil. This indicates that bubble reaching and incivility are moderated by the national political context. Our results complicate the simple hypothesis of a universal impact of neutral bubble reachers across contexts.

## Introduction

Theorists have long recognized advances in communication technologies as double-edged swords. Over 2000 years ago, Plato worried about books replacing the experience of face-to-face conversation, even as he pioneered a novel genre of writing—the philosophical dialogue—designed to capture and disseminate that experience in a new medium [1, 2]. The printing press led to greater literacy, increased access to news and learning, and to the efflorescence of

**Funding:** TS and JK - project SocialNet (process 2023/00148-0 from Sao Paulo Research Foundation - FAPESP) TS - CNPq (grants 314603/2023-9 and 441444/2023-7). The funders had no role in study design, data collection and analysis, decision to publish, or preparation of the manuscript.

**Competing interests:** The authors have declared that no competing interests exist.

highly partisan pamphleteering marked by aggressive attacks on enemies with little regard for truth or civility.

Observers of social media have noticed similar tensions. Early optimism saw opportunities for a renewal of democratic discourse, with individuals from diverse backgrounds finding possibilities to learn from and interact with others different from themselves [3, 4]. In this view, exposure to divergent views from members of outgroups would promote cross-cutting ties and thereby encourage mutual understanding, civility, and inclusivity. The Internet could be a school for deliberative democracy [5, 6].

This optimism quickly waned as online communication came to resemble the very fragmentation and balkanization that it had promised to overcome. Instead of cross-cutting ties, social media seemed to breed ideological homophily marked by "filter bubbles" or "echo chambers" [7, 8]. These, to some scholars, evoked the forms of interaction that arise when bridging ties are absent [9, 10]. By insulating partisans from countervailing views, existing viewpoints are intensified, and alternative ideas are delegitimized, potentially fraying the fabric of democratic society.

The metaphor of "echo chambers" [11] refers to epistemic environments in which individuals or groups interact with like-minded people or sources that reinforce their existing beliefs or opinions. In an echo chamber, information, ideas, and opinions are "echoed" and "amplified" within a closed network, leading to a reinforcement of existing viewpoints and a lack of exposure to diverse perspectives [11]. While the concept of "filter bubbles" refers to the algorithmic filtering that selectively curates content based on a user's past behaviour, preferences, and demographic information [12]. In a filter bubble, users are presented with information that aligns with their interests and beliefs while filtering out opposing viewpoints or dissenting opinions [12].

A common response to the sense of fragmentation has been to encourage exposure to more cross-partisan sources of information [13, 14]. By offering citizens certified facts from neutral parties, platforms seek to reach across "bubbles" and "echo-chambers" to create a common basis for generating a meaningful conversation. However, recent research has largely popped the bubble on the ideological bubble perspective [15, 16]. In fact, much evidence indicates that online networks offer high exposure to diverse viewpoints and news sources and that users proactively seek out those with divergent views [17, 18]. They do so not necessarily to converse, however, but to engage the other in ways that do not moderate but reinforce prior commitments [19].

To examine these dynamics, we extend Kobellarz et al. [20], which investigated the role of central Twitter users, whom they characterized as *"bubble reachers."* Bubble reachers are users who distribute content that reaches other users with diverse political opinions [20]. In order to identify these central users, the authors created a novel centrality metric called intergroup bridging [20]. By studying Twitter discourse during the 2018 Brazilian and 2019 Canadian elections, they identified bubble reachers such as @UOLNoticias in Brazil and @globalnews in Canada [20]. While such accounts disseminate content that engages ideologically diverse audiences, users nevertheless seem to respond to that content in strongly partisan ways: they share content that aligns with their own partisan orientation. This same dynamic held in both the Brazilian and Canadian contexts, despite the very different levels of political polarization in those countries.

The present article builds upon this research to study further unexplored properties of content shared by bubble reachers, specifically the quality of conversations and comments that it generates. Cross-cutting ties do not necessarily mitigate the tendency to maintain ideological groupings via selective reinforcement of existing views. Still, a question remains as to the quality of discussion that bubble reachers support. Do they offer spaces that mitigate rancorous

discourse by providing a common starting point for conversation, even if that starting point is differently interpreted? Or do they provide the discursive equivalent of "combat zones," gladiatorial arenas where partisans meet not to deliberate and converse, but to fight?

Bubble reachers may intervene in civil discourse in different ways: some bubble reachers may "open the hand" for broader conversations in a relatively neutral way, while others "raise the fist" and offer not mutual understanding but mutual antagonism. But to what end? How bubble reachers navigate the civil sphere and the public reactions they elicit occur within a wider cultural context. Particularly relevant is the rise of populism and political polarization: relentless challenges against media institutions by populist leaders, epitomized by "fake news" narratives, have eroded trust in these institutions and credibility in their performances of neutrality [21]. In these contexts, a neutral bubble reacher may try to "open the hand" for broader conversation, but still receive the fist. As the room for neutrality narrows, even ostensibly non-political content may elicit antagonistic discourse as it becomes filtered through a partisan lens. Overall, whether bubble reachers support more or less hostile discursive spaces may not only vary by their styles of intervention, but also by the cultural context in which their performances are received.

Addressing these issues is the central goal of this paper: we examine how ideologically neutral bubble reachers differ from ideologically partisan accounts in the level of uncivil discourse they provoke, and explore how this varies in two contexts where populism is more and less prominent in political culture. We do that by examining the contexts of Brazil, where populism has deep roots in the political culture, and Canada, where the political culture is comparatively moderate. By analyzing the communication style provoked by neutral bubble reachers and ideologically partisan accounts in these distinct political cultures, this study marks a substantial step forward regarding past research.

The rest of this study is organized as follows. The section "Related Work and Hypotheses" presents a comprehensive literature review and related hypothesis about uncivil discourse and brokerage processes in online media, the relationship between the political context and neutrality, an overview of the political context of the two studied countries (Brazil and Canada), and methods for incivility detection in online discourse and its underlying challenges. The section "Materials and Methods" presents the data sources and methods we use to evaluate the hypotheses, including the measure used to identify bubble reachers, a presentation of the datasets, the text pre-processing method applied for toxicity inference from these datasets (using existing methods for inferring "toxicity" as a proxy for incivility), and methods for hypothesis testing. The "Results" section presents and discusses the results regarding our hypotheses. Finally, "Discussion and Conclusion" concludes the study and presents potential limitations and future work. Supporting information reports additional sensitivity analyses.

The pre-processed dataset and all the code necessary to replicate the statistical tests carried out in this research were made publicly available on the project website: https://sites.google.com/view/onlinepolarization [22]. Disclaimer: This file includes words or language that is considered profane, vulgar or offensive by some readers. Due to the topic studied in this article, quoting offensive language is academically justified, but we nor PLOS in no way endorse the use of these words or the content of the quotes. Likewise, the quotes do not represent the opinions of us or that of PLOS, and we condemn online harassment and offensive language.

## Related work and hypotheses

### Uncivil discourse and brokerage in online media

Scholars often evaluate discourse in the public sphere using the Habermasian normative ideal of deliberation based on reason-giving, reciprocity, and civility as a benchmark [23]. Applied

to the online sphere, discourse arguably falls short of this standard: rather than a site for civil deliberation, social media resembles a rancorous town square, where uncivil discourse arguably holds sway [21]. Whereas civility involves mutual respect, incivility violates these norms [24, 25]. Uncivil discourse is not necessarily incompatible with deliberative democracy, to be sure [26]. Heated debates naturally involve aggressive and conflictual discourse, and studies find that this can make discussions more engaging while still appealing to deliberative persuasion [24, 27, 28]. Be that as it may, uncivil discourse is regarded as both symptomatic of polarization, and as a fuel that exacerbates it further [29, 30]. Research finds that uncivil discourse can elicit a feedback loop that provokes it in others, that alienates more civil participants from platforms, and distorts perceptions of political life altogether [31, 32]. While it may not be inherently harmful to deliberative political talk, it is often deemed problematic to the extent that it contributes to polarization and intolerance [26].

Concerns over the proliferation of uncivil discourse have motivated inquiry into the conditions that underlie it. One approach emphasizes fragmentation in the media ecology [8, 33, 34]. Social media is putatively balkanized into "echo-chambers", which amplify a set of in-group beliefs, discredit alternative viewpoints, and discourage charitable engagement with outsiders [35]. Recent work suggests fragmentation is not primarily about epistemic *closure* in so far as members are still exposed to outside information [15], but rather epistemic *discrediting* where outside information is readily dismissed [19]. According to Bright et al. [36], echo chambers thrive on steady exposure to low levels of oppositional views—they rely on conflict to enliven group identity, while providing cognitive tools to manage dissonance in ways that reinforce extreme belief. Fragmentation of this sort can set the backdrop for uncivil discourse. Segmented subcultures produce group identities with norms of civility that exclude outgroups, thereby licensing disparaging remarks against them as occasions to signal in-group affiliation [37]. While this can vary by communication norms in different communities [38] scholars also identify platform characteristics, whether user anonymity, homophily, or recommendation algorithms favouring intense emotional engagement, that undergird these dynamics, informing structural conditions that promote the expression of uncivil discourse [23, 39, 40].

But if uncivil discourse stems from fragmentation in the online public sphere, it also requires meeting points. "Bubble reachers"—central brokers that form bridges between opposing bubbles—can occasion the intersection of opposing partisans [20]. The term "bubble reachers" grows out of the computer science literature and refers to users on social media platforms who distribute content that reaches users with diverse political opinions, possibly bridging ideological divides and facilitating cross-partisan conversations [20]. They play a role in countering the phenomenon of "filter bubbles" [12] or "echo chambers," [41] where individuals are exposed more frequently to information and perspectives that reinforce their prior beliefs. To understand the concept of bubble reachers in a broader network science context, we can draw parallels with the terms "brokerage" and "brokers." In network science, brokerage refers to the role played by individuals or entities that connect otherwise disconnected groups or individuals within a network [42]. They occupy positions that span structural holes [42], which are gaps between distinct clusters or communities within a network. By spanning the structural holes, brokers create weak ties [43], acting as intermediaries and facilitating the flow of information, resources, or relationships between different parts of the network. Bubble reachers, in this sense, can be seen as performing a form of brokerage between partisan groups in a polarized context. They bridge ideological gaps and act as connectors between users with diverse political opinions, similar to how brokers bridge gaps between different groups or clusters in a network. Research suggests that, in practice, legacy media outlets are important bubble reachers in this regard [20, 44, 45]. For instance, Magin et al. [45] show that legacy media

outlets help politically extreme individuals sustain engagement with the fundamental essence of public dialogue.

How bubble reachers relate to civil discourse is unclear and variable, however. By closing the distance between opposing groups, they open space for confrontation where greater incivility might be expected. At the same time, bubble reachers may also engage alternative perspectives out of a pluralist commitment to pursue neutral dissemination of information and uphold trust in a diverse public [46]. To the extent that bubble reachers hold these commitments and are successful in their ambitions, exchange between opposing partisans may abide by norms of civility. It follows that neutral bubble reachers—those central nodes that avoid a partisan tilt—may, in some circumstances, temper the proliferation of uncivil discourse compared to those with a clear partisan valence. While this approach does not pre-determine what kind of media outlet emerges either as a bubble reacher in general or a neutral bubble reacher in particular, empirically neutral bubble reachers tend to be legacy media outlets [20].

These insights about the nature and types of bubble reachers inform the first hypothesis we examine in this paper:

**H1: Neutral bubble reachers will tend to reduce uncivil discourse relative to partisan accounts**.

## Political context and performances of neutrality

Scholarly focus on the role of platform characteristics and the structure of online networks tends to elide the role of differences in political culture. However, in the case of bubble reachers, there is good reason to believe their relation to uncivil discourse may vary across cultural contexts. Alexander [47] conceives of democratic life as consisting of symbolic performances that are interpreted as compelling or not based on their wider resonance with cultural structures. The appeal to principles of objectivity and fairness by neutral bubble reachers in their rendering of current affairs can likewise be interpreted as a performance of neutrality that the public assesses, and assigns varying degrees of credibility. Media brokers of this sort provide a crucial medium for how the public derives an understanding of world events and opinions on the state of affairs [48]. In contexts of consensus, the public assign credibility to the neutrality of bubble reachers, allowing them to provide what Talcott Parsons [49] referred to as a shared "definition of the situation" grounding conflict in common criteria for validity. When these claims lose their credibility, however, the civil sphere runs the risk of spiralling into crisis.

Against this backdrop, the rise of populism reflects a challenge to the social fabric: it is both a response to enervated mediating institutions and also a force working against the possibility of repair. Recent scholarship understands populism as a style or form of political communication which can be directed to many goals and ideological aims [50–56]. Although different in emphasis, these approaches agree that "the populist script" [54] tends to revolve around two core axes: a vertical one in which some "elites" are contrasted to "the people," and a horizontal one, in which insiders are pitted against outsiders. Establishment media is often framed along the vertical axis as "elites" out of touch with the day-to-day concerns of "ordinary people". Within this framework, universalistic ambitions of bubble reachers can falter. As Alexander [57] observes, a common trope in many populist scripts involves the critique of impartial expertise, in which experts' claims to neutrality are reinterpreted as partisan defenses of "establishment" interests that run counter to those of the "common people". Consistent with this, populist citizens are found to subscribe to a "false consensus", assuming that their opinions are overwhelmingly shared by others, but unfairly maligned by hostile and biased mainstream

media [58]. Performances of civility lose credibility in these contexts. Distinguished by its combative and morally-charged style, populist frames amplify authenticity over civility as a criterion for evaluation [59]. Careful and restrained forms of civil discourse fall short of these standards of expressivity, appearing weak and ambivalent by comparison [60].

Populism works in conjunction with polarization. As the legitimacy of mediating institutions recedes and neutral discursive spaces contract, the ambit of partisanship presumably expands—even non-political domains gradually become assimilated into partisan conflict. The "oil-spill" model of polarization, for instance, documents how partisanship can permeate even areas of social life which are seemingly apolitical and inconsequential [61, 62]. This creates an additional reason why neutral bubble reachers may struggle to uphold spaces for civil discourse in a more polarized context. Significant scholarly literature suggests political discourse is especially rancorous. In contrast to news concerning science, technology, weather, art, and culture, Salminen et al. [63] find that topics with a political valence provoke more toxic comments. However, to the extent that polarization interfuses social life broadly, giving even non-political content a partisan valence, the distinction between the political and non-political may become blurred. As a result, in more polarized settings, uncivil discourse may be expected across different topics, jeopardizing neutral bubble reachers' ability to leverage non-political topics as meeting points for civility.

In these ways, contexts where political culture is characterized by populism and polarization, may affect uncivil discourse and the relation of neutral bubble reachers to it. Uncivil discourse may be expected to be more pronounced in these public spheres, and the role of neutral bubble reachers in curbing it may falter. Rather than tempering discourse, these actors may even elicit greater invective. Their claim to objectivity and fairness may be reframed as a veiled cover for the interests of the "establishment" and provoke greater scorn as a result [58]. Furthermore, as the space for neutrality contracts, uncivil discourse may be expected to enter even seemingly non-political domains. These considerations lead to three additional hypotheses:

**H2: Uncivil discourse will be more pronounced in contexts that are more populist and politically polarized**.

**H3: Neutral bubble reachers will produce greater uncivil discourse than partisan accounts in contexts that are more populist and polarized**.

**H4: Uncivil discourse will be evident in both political and non-political topics in contexts that are more populist and polarized**.

### Populism and polarization in Canada and Brazil

We examine the role of context by considering how bubble reachers operate differently in Canada and Brazil. Populism is not foreign to either context, but has deeper roots in Brazil. While populism has had a significant steady influence on Brazil's politics on both the left and the right since the early 20th century, its role in Canadian politics has been small by comparison. Lipset [64] famously enshrined the image of Canada as a country of political moderation when he contrasted the populist and anti-statist strains in American political culture to Canadian political culture, which was distinguished by its greater deference to elites and government authority. Some more recent scholarship disputes this characterization and suggests both populism and polarization are on the rise in Canada [65–67]. Nevertheless, studies still observe greater recent levels of affective polarization in Brazil as compared to Canada, and greater support for political leaders employing populism more generally [68–70]. These differences are

especially pronounced in Canadian national electoral politics, where populism remains marginal compared to Brazil. To be sure, populist leaders have emerged in Canadian municipal politics [56]. But whereas Canadian populist parties like the People's Party of Canada have achieved little electoral success, both right-wing and left-wing populism have found success in Brazil at the national level. Jair Bolsonaro's election in 2018, for instance, was notably fuelled by an anti-establishment campaign which frequently targeted the media class as purveyors of "fake news." Scholars observed the important role of digital media in the culmination of these events. Digital media became a major conduit for anti-establishment news, and served to normalize Bolsonaro's populist discourse [71–73]. Communication campaigns mobilized citizens to engage in polarized debates, with platforms like Facebook and Twitter becoming the site of rancorous and even violent discourse.

Despite these differences, Canada and Brazil share important commonalities. Both are large federal systems and constitutional democracies, containing populations that are multicultural and racially diverse. Importantly, citizens in both countries are active on social media at roughly similar rates. Recent reports find that Facebook's ad reach is equivalent to 50.5% and 53.5% of the total population, with 61.1% and 61.7% of the eligible population using Facebook in Brazil and Canada respectively. Given the different levels of populism and polarization in both countries, these structural similarities make Brazil and Canada appropriate cases for comparative analysis [74, 75]. They allow us to assess our hypotheses on how bubble reachers may operate differently in contexts where populism and polarization are more prominent and have deeper roots in political culture (Brazil), compared to contexts where political culture has historically been more moderate, and populism and polarization are less pronounced (Canada), while holding important structural variables constant.

## Detecting uncivil discourse in Canada and Brazil

While the perception of incivility in discourse is to some degree a subjective matter, the research literature identifies some characteristic features that demarcate incivility, such as *ad hominem* attacks (direct attacks on the person instead of their arguments), vulgarity and exaggerations [76]. In short, incivility involves communication that violates norms of courtesy and respect in social interactions, especially in the political sphere, where discussion of ideas and opinions can be heated. In the online sphere, incivility theoretically overlaps with "toxic discourse", which refers to disrespectful or irrational comments that may lead users to leave discussions [77]. Our empirical analysis detects incivility by engaging with work on toxic discourse.

To provide context, these are examples of web comments obtained from one of the datasets studied in this article, which were manually annotated by three independent evaluators regarding their toxicity [78], demonstrating uncivil discourse: "tirso, you drunk, I'm not just a man on the internet, come here or give me your address if you're a man, and I'll come and talk to you in person. My address is [. . .], (it's very close to the current presidency of your luladrão boss) come here and talk this nonsense" and "KKKKKK now DataFAKE does a minimally truthful survey, because it was already getting bad. kkkkkkkkk ridiculous! They tried to manipulate the Brazilian vote but this is over!! they just lost the rest of the credibility they had, if they had any!". Meanwhile, these are examples of comments also manually annotated demonstrating civilized discourse [78]: "In the debate for governor of Rio held on Globo on the 25th, Eduardo Paes scored a great victory, he deserves to be elected." and "Bolsonaro did not claim medical recommendation, the doctors themselves gave an interview recommending that he not go. Everyone saw it."

Researchers have developed several methods to identify such comments, ranging from keyword-based proposals [79] to strategies that use machine learning [80–85]. Given the speed with which discussions are growing on the Web [86], different models for large-scale toxicity identification were created to solve specific challenges, such as for dealing with online comments [83, 85]. Among the challenges is the fact that this type of text contains varying degrees of subtlety inherent to the language, cultural aspects, specificities of context, presence of sarcasm, and use of figures of speech, which can make the actual toxicity of a comment ambiguous. Other challenges include the fact that online comments are usually short texts, often containing spelling errors that occur sparsely in the dataset [83], making it challenging to create models with a generalized capacity to identify uncivil instances. Likewise, machine learning-based approaches need considerable quantities of high-quality annotated data, which can present questions concerning model training [87]. Thus, we can find several studies that concentrate on studying and criticizing toxicity classification methods [87, 88]. While "toxicity" and "incivility" may not perfectly overlap, the existing tools for measuring toxicity are highly developed and provide a useful proxy for online incivility. Future research may benefit from developing direct measurements of incivility purpose-built for that task.

Uncivil discourse is also sensitive to context. For example, expressing hostility about the indigenous community in Brazil will likely be different in the context of a pro-Bolsonaro discussion and a discussion in the context of Funai (Brazilian governmental protection agency for Amerindian culture and their interests) supporters. Uncivil discourse also varies depending on target groups—for instance, expressions employed to express hate against a community in Brazil are distinct from expressions employed against Latin Americans in the United States. These challenges are even greater when it is necessary to classify and compare comments in different languages [83]. It is not uncommon for text analysis tools to work only for English or to have more resources available for this language, since it has been the subject of more training instances and research [89–91]. Given this situation, researchers often translate non-English content into English before performing automated text analysis tasks. Kobellarz et al. [78], however, found that while this approach can achieve good performance in some domains, such as sentiment analysis [90], performance for toxicity scores is poor. They, therefore, considered a translated version of a Brazilian Portuguese comments dataset to identify whether the original or the translated version would be more suitable for the study [78]. To infer toxicity scores in online comments, the authors used the Perspective API (https://perspectiveapi.com), a multilingual model widely employed for this task [63, 92–95]. They verified that the translation process artificially reduced the overall toxicity scores of the datasets by penalizing highly toxic comments in their original language [78]. The present study follows this guideline, keeping the datasets in their original language to infer toxicity.

## Materials and methods

To evaluate our hypotheses, we followed a method for comparing the toxicity scores in multiple comment datasets from distinct sources and languages. This involved a structured process of five sequential steps:

1. Apply a centrality metric to identify bubble reaching accounts.

2. Obtain the necessary data to study our hypothesis.

3. Perform essential standard text pre-processing steps.

4. Infer incivility levels from the studied comments.

5. Evaluate each hypothesis.

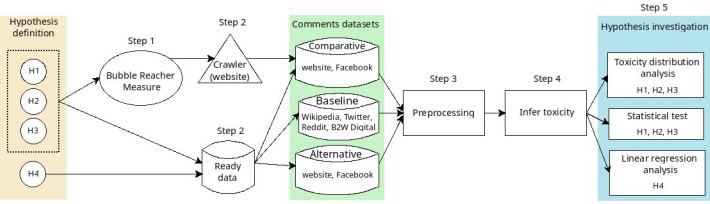

**Fig 1. Overview of steps to answer our hypotheses.**

Fig 1 summarizes these steps, which are explained in the following sections.

## Bubble reacher measure

The first crucial step to evaluating our hypotheses was to identify bubble reachers, central users who distribute content that reaches other users with diverse political opinions. To do so, we relied on the results of Kobellarz et al. [20], where the authors studied the role of these central users during the 2018 Brazilian and 2019 Canadian elections. For that, understanding the political context surrounding each election is fundamental for a comprehensive analysis. In the case of Brazil, during the election period, the polls (http://media.folha.uol.com.br/datafolha/2018/10/19/692a4086c399805ae503454cf8cd0d36IV.pdf) revealed a notable degree of polarization among voters. The electoral contest revolved around two primary candidates: Jair Messias Bolsonaro, who symbolized the potential for a 15-year departure from the ruling Workers Party (PT), and Fernando Haddad, who represented the continuity of PT's governance. This situation arose after a brief period during which a PT president elected was impeached and replaced by her vice-president. Ultimately, Bolsonaro emerged victorious with a majority of 55.13% of the valid votes, while Haddad secured 44.87% of the votes in his favor (https://politica.estadao.com.br/eleicoes/2018/cobertura-votacao-apuracao/segundo-turno). In Canada, Justin Trudeau represented the Liberals, which had previously held a parliamentary majority after unseating the Conservatives in 2015. In a close election, the Liberals won 157 (39.47%) seats in parliament, while the Conservatives, led by Andrew Scheer, won 121 (31.89%) (https://newsinteractives.cbc.ca/elections/federal/2019/results). The (left-wing) New Democratic Party continued to lose ground from its 2011 peak, especially in French Quebec, where the Liberals and the separatist Bloc Québécois subsequently gained ground. The 2019 election resulted in the Liberals forming a minority government, which has historically exhibited instability, since the prime minister relies on representatives of other parties to remain in power (https://web.archive.org/web/20130627154515/http://www2.parl.gc.ca/Parlinfo/compilations/parliament/DurationMinorityGovernment.aspx). Despite Canada's multi-party political system, national politics has predominantly revolved around the traditional left-right ideological differences since at least the 1980s [96].

To measure and find bubble reachers, Kobellarz et al. [20] developed a new centrality measure called intergroup bridging, which was inspired by the "bridgenness'" algorithm [97], adapted from Brandes "faster algorithm" to calculate betweenness centrality [98]. This novel centrality measure demonstrated a significant improvement in identifying global bridges in a polarized network, highlighting users with a greater ability to distribute content to polarized bubbles with diverse political opinions.

Kobellarz et al. [20] used the intergroup bridging centrality metric to identify the most central users in *retweet* networks in the Canadian and Brazilian political situations. From the results, the 100 accounts with the highest intergroup bridging centrality value were selected.

After that, the links from messages retweeted by other users from these bubble reaching accounts were extracted, as well as the respective domains and latent topics present in the contents pointed out by these links. In the last step, they identified the political polarity of the entities (contents, domains, and topics) using a metric that the authors called $RP(H)$—relative polarity—which is a measure obtained from the weighted average of the polarity of the users who retweeted a certain entity. The $RP(H)$ metric, in short, expresses the political bias of an entity concerning the audience from which it generates engagement. The result of $RP(H)$ is a value in the continuous range [−1.0; + 1.0]. Positive values represent a right-wing political orientation, negative values represent a left-wing orientation, and values close to 0.0 represent a neutral orientation. Based on $RP(H)$ value, entities were labelled according to their political orientation: the first third of values on the $RP(H)$ scale represents the left-wing (L) users, with $RP(H) \in [−1; −1/3[$, the second third, the neutral users (N), with $RP(H) \in [−1/3; 1/3]$ and the last third, the right-wing users (R), with $RP(H) \in ]1/3; 1]$.

In the next section, we provide an explanation of how the intergroup bridging and $RP(H)$ metrics were used to create two of the datasets studied in this research.

## Comments datasets

**Comparative datasets (for H1, H2 and H3).** To assess the H1, H2, and H3 hypotheses, we first obtained representative comments made on content published by *neutral bubble reachers* and *partisan accounts*. We, therefore, generated four datasets, two to represent neutral bubble reachers comments, one in Brazilian Portuguese and the other in English, and two to represent partisan accounts comments, also one in Brazilian Portuguese and the other in English.

To capture neutral bubble reachers' comments, we relied on [20]. For the present study, we obtained comments made about news articles published by the top 3 bubble reaching domains with neutral $RP(H)$ in each country. To simplify the understanding of which dataset we refer to in the next sections, we've labelled Brazil's dataset NEUTRAL_REACHER_pt and Canada's dataset NEUTRAL_REACHER_en—the prefix NEUTRAL_REACHER means "neutral bubble reacher" to facilitate the identification of these specific cases on further analysis. For each domain in these datasets, we created a web scraper to capture all comments from news articles published during the electoral period in each country.

Considering that all comments in NEUTRAL_REACHER_pt and NEUTRAL_REACHER_en were made on content produced by neutral bubble reachers, we also collected comments made on contents produced by partisan accounts to enable us to compare these distinct cases. Following the recommendations of [78] we kept the comments in their original language, even in comparisons between different languages [78]. We, therefore, collected two additional datasets to be compared to NEUTRAL_REACHER_pt and NEUTRAL_REACHER_en, one in Portuguese and the other in English, respectively, containing comments on content distributed by partisan accounts. To ensure a representative sample from these accounts, we identified users among the top 100 bubble reachers in the Canadian and Brazilian datasets [20] with partisan $RP(H)$ scores ($RP(H) < −0.333$ or $RP(H) > +0.333$)—those who act as bridges between bubbles but are more likely to generate engagement on one side of the political spectrum, given by their $RP(H)$ value.

Unlike what was possible to do with neutral bubble reachers to collect comments on news articles on their respective websites, it was not possible to replicate this same step for the case of partisan accounts, since among the top 100 bubble reachers with partisan $RP(H)$, there were few linked to sites with comments sections and none with a considerable number of comments. Therefore, alternative sources of comment data on partisan news media pages on Facebook in

Portuguese and English were obtained to compare with the NEUTRAL_REACHER_pt and NEUTRAL_REACHER_en datasets, respectively.

For the English case, we cross-referenced the top 100 bubble reachers with partisan $RP(H)$ in Canada with a Facebook dataset containing comments on 20, 000 posts made by American news organizations and personalities [99]. This dataset has been labelled as PARTISAN_REACHER_en. The American political culture is recognized for its polarization [100, 101], and some evidence shows that it is higher than the Canadian scenario [69]. Despite differences in political culture, Canadians also engage with these sources, albeit to varying degrees. Thus, we selected this dataset as an approximation for partisan accounts for the Canadian situation.

However, for the Portuguese case, it was not possible to cross-reference this same Facebook dataset with any bubble reacher with partisan $RP(H)$ like what was possible for PARTISAN_REACHER_en, so we obtained another Facebook dataset containing comments on Brazilian Conservative pages from 2012-01-01 to 2018-12-31, which can be considered partisan representative accounts (mostly right-wing) [102]. This dataset has been labeled as PARTISAN_pt. Note the lack of the "reacher" in the name of this dataset: it was removed intentionally, because it does not include comments from cross-referenced bubble reaching accounts, in contrast to what was possible with other datasets.

In summary, here are the datasets introduced in this section:

**NEUTRAL_REACHER_pt:** this dataset comprises comments captured from news media websites representing these top three neutral bubble reaching domains in the Brazilian election setting: noticias.uol.com.br, g1.globo.com, and extra.globo.com.

**NEUTRAL_REACHER_en:** this dataset comprises comments captured from news media websites representing these top three neutral bubble reachers in the Canadian election setting: cbc.ca, globalnews.ca, and theglobeandmail.com.

**PARTISAN_pt:** This dataset comprises comments captured from Brazilian conservative Facebook pages (mostly right-wing), including: padrepaulo, flaviobolsonaro, CampanhadoArmamento, carvalho.olavo, OPesadelodeQualquerPolitico2.0 and MisesBrasil. It is worth mentioning that comments in this dataset are not linked to the specific electoral contexts in Brazil or Canada, like the ones obtained to represent neutral bubble reachers. Also, these pages can't be considered bubble reaching sources, like what was possible with other datasets with "reacher" in their name.

**PARTISAN_REACHER_en:** this dataset comprises comments captured from Facebook news pages representing the partisan accounts (left-wing or lean toward the left according to the All Sides Media Bias Report (the tool assigns a rating of Left, Lean Left, Center, Lean Right, or Right to each media source. The ratings reflect the average view of people across the political spectrum, obtained by exploring a scientific, multipartisan analysis. More details about the method can be found in: https://www.allsides.com/media-bias/media-bias-rating-methods— The All Sides Media Bias report was obtained on Dec 12, 2022): bloomberg_politics, huffington_post, ny_times. It is also important to note that comments in this dataset are not linked to national elections in Brazil or Canada.

We call this set of datasets "Comparative datasets."

**Reference datasets.**   As our hypotheses concern comparisons regarding incivility, we need a method to identify and compare the incivility of comments in the neutral bubble reachers (NEUTRAL_REACHER_pt and NEUTRAL_REACHER_en) and partisan accounts (PARTISAN_pt and PARTISAN_REACHER_en datasets). To do so, we utilize four reference datasets; two of them composed of highly uncivil comments, one in English (UNCIVIL_en) and another in Brazilian Portuguese (UNCIVIL_pt), and the other two containing more civil comments, also one in English (CIVIL_en) and another in Brazilian Portuguese (CIVIL_pt). These reference datasets are presented below.

**UNCIVIL_en**: contains highly uncivil United States English comments (not translated) taken from an open dataset with human-labelled Wikipedia comments according to different categories of toxic behaviour (https://www.kaggle.com/competitions/jigsaw-toxic-comment-classification-challenge/data), including "toxic", "severe_toxic", "obscene", "threat", "insult", and "identity_hate". The variable "toxic", representing the degree of toxicity of the comment, was used to select 5, 000 comments with the highest value for this metric and compose the UNCIVIL_en dataset. Examples of comments can be found on the project's website (https://sites.google.com/view/onlinepolarization). Disclaimer: This file includes words or language that is considered profane, vulgar or offensive by some readers. Due to the topic studied in this article, quoting offensive language is academically justified but we nor PLOS in no way endorse the use of these words or the content of the quotes. Likewise, the quotes do not represent the opinions of us or that of PLOS, and we condemn online harassment and offensive language.

**UNCIVIL_pt**: contains highly uncivil comments in Brazilian Portuguese obtained from tweets manually annotated according to different toxicity categories in a publicly available dataset [103]. The available categories were "non-toxic", "LGBTQ + phobia", "obscene", "insult", "racism", "misogyny", and "xenophobia" [103]. To select a representative sample, the number of toxicity categories linked to each comment was counted, except for the "non-toxic" category. This count was used to select 5, 000 comments with the highest amount of toxic categories to compose the UNCIVIL_pt dataset. Examples of this dataset can be found on the project's website (https://sites.google.com/view/onlinepolarization). Disclaimer: This file includes words or language that is considered profane, vulgar or offensive by some readers. Due to the topic studied in this article, quoting offensive language is academically justified but we nor PLOS in no way endorse the use of these words or the content of the quotes. Likewise, the quotes do not represent the opinions of us or that of PLOS, and we condemn online harassment and offensive language.

**CIVIL_en**: contains more civil comments in United States English (not translated) obtained through Reddit's public API, a network of communities where people with common interests interact in a forum-like system. To collect the data, the most popular 100 posts from the communities (*subreddits*) \AskHistorians, \changemyview, \COVID19, \everythingScience, and \science were selected. These communities were chosen because they contain potentially more civil discussions. Indeed, inspection of a sample of responses showed a high level of civility and earnestness. These features are likely due to the fact that stricter rules for posting were explicitly informed and seemed to be followed by their participants. Thus, this dataset was composed mostly of constructive comments. In addition to the text of the comments, other attributes were obtained, among them the "score" of the comment, which is a metric calculated by subtracting the negative votes from the positive votes that a given comment received. This metric was used to select 5, 000 comments with the highest score to compose the CIVIL_en dataset. Some comments examples in this dataset are: "According to a paper published in IEEE Transactions on Computational Social Systems by researchers at The University of Notre Dame, some 73 percent of posts on Reddit are voted on by users that haven't actually clicked through to view the content being rated. Hopefully, this information allows 3 out of 4 people to not have to read through the article." and "I have lived with the prospect of an early death for the last 49 years. I'm not afraid of death, but I'm in no hurry to die. I have so much I want to do first.".

**CIVIL_pt:** composed of more civil comments in Brazilian Portuguese obtained from a database of product reviews in a famous e-commerce business, B2W Digital, responsible for the *americanas.com* website, whose data were obtained between January and May 2018 [104]. This dataset was selected considering that among positive and lengthy product reviews, there

would be a less hostile and uncivil discussion. Therefore, during data cleaning, evaluations with a score lower than 5 and that contained less than 20 unique characters were eliminated. This is important because the incidence of texts with repeated words was identified in cases where the evaluator only filled in the text field with no intention of making a careful assessment. After cleaning, the longest 5, 000 evaluations were selected to compose the CIVIL_pt dataset. Some comments examples (translated to English) on this dataset are: "I had researched the product previously and it met my expectations, despite the design flaws that I was already aware of. Furthermore, the product arrived very quickly and in perfect condition." and "The table is light and small, as the name suggests. It has an excellent texture and the pen feels comfortable in the hand. The settings are in a very intuitive interface and overall allow us to have excellent adjustments for use. Excellent for beginners who want to experiment with digital art.".

We call this set of datasets "Reference datasets."

**Alternative datasets (for H4 and sensitivity analysis).** In politically polarized and populist environments like Brazil, even neutral content may acquire a political connotation. This raises the question of whether ideologically neutral bubble reachers behave differently in such contexts. H4 delves into this possibility by examining how the incivility of comments varies between political and non-political content in contexts that are populist and polarized, like in Brazil. To analyze this, we gathered discussions from ideologically neutral bubble reachers that could be distinguished based on whether they specifically addressed political topics or not. For this purpose, comments were obtained from the Globo G1 [105] and the New York Times (https://www.kaggle.com/datasets/aashita/nyt-comments) news websites. Both websites organize comments by topic, enabling us to identify which comments explicitly referenced political or non-political topics. This categorization allows us to examine incivility in political and non-political content in the Brazilian populist and polarized context using Globo G1 data, and compare it with the American context using New York Times data. These datasets, labelled as G1_SITE_pt and NYT_SITE_en, respectively, are presented below.

**G1_SITE_pt:** composed of comments written in Brazilian Portuguese on news articles published on the Globo G1 (g1.globo.com) website between March 28, 2020, and November 11, 2020 [105]. It comprises a total of 1, 059, 672 comments spanning 18, 014 news articles [105]. Considering that g1.globo.com domain is also present in NEUTRAL_REACHER_pt dataset, but contains comments in distinct time periods, it also enables an investigation into potential variations in toxicity levels between comments made in a political context, as captured in NEUTRAL_REACHER_pt, and those made in other context, as captured in G1_SITE_pt.

**NYT_SITE_en:** composed of comments obtained from the New York Times website during March 2018 (https://www.kaggle.com/datasets/aashita/nyt-comments), from which were selected 5, 000 random comments from political sections and 5, 000 random comments from non-political sections.

Additional data was collected to address limitations regarding comparative datasets, which are presented in S1 Appendix. We call this set of datasets "Alternative datasets."

S2 Appendix. presents a summary of all datasets and their corresponding sources for Comparative, Reference, and Alternative groups for quick reference.

**Note on using Facebook as a data source.** It is important to consider the validity of using Facebook data, considering that neutral bubble reachers were identified from Twitter data [20]. First, the neutral bubble reachers that emerged from the Twitter analysis were major legacy media outlets. That these outlets were central in reaching across ideological bubbles is consistent with other work likewise finding evidence that legacy outlets have the greatest audience overlap in consumption of digital news [44]. This makes us confident that the neutral bubble reachers identified using Twitter data should characterize these outlets more broadly,

including on Facebook. Second, incorporating the Facebook dataset allows for greater information diversity in the analysis of comments. Relying exclusively on Twitter as a data source would lead to results that are skewed toward Twitter platform practices and the political representativeness of its users. Research by Barbera and Rivero has shown that Twitter users who engage in political discussions tend to be male, reside in urban areas, and possess extreme ideological preferences [106]. Moreover, Twitter's character limit constrains discourse complexity inhibits reflexivity and tends to encourage uncivil behaviour due to the impersonal nature of the platform [107]. By including data from Facebook, we can broaden our understanding of the phenomenon across multiple platforms. Furthermore, previous studies on polarization and hate speech have heavily relied on Twitter data alone [41, 108, 109]. Our research aims to expand the scope by extrapolating our findings to other platforms, enabling a more comprehensive examination of the phenomenon.

## Pre-processing

The pre-processing step was performed to maintain maximum integrity in the datasets presented in the previous sections, only removing noisy instances that could negatively impact performance. S3 Appendix presents pre-processing details.

## Inferring toxicity as a proxy for incivility

A widely used model for identifying toxicity in academia and industry is the Perspective API [77], which is a multilingual tool available for free through a Google initiative called Perspective API (https://perspectiveapi.com). This model can be accessed through a public API, which allows users to identify different toxicity types in online comments. Given that this is a widely adopted tool by major news outlets for moderating comments on their portals, in addition to the fact that it is openly available, having support and good performance on different languages [78], Perspective API was chosen to be applied to this research. Previous studies indicate that Perspective API satisfactorily captures the toxicity of social media content [93–95]. For instance, Rajadesingan et al. [93] show that its performance in identifying toxicity is comparable to toxicity labelled by humans. Some studies suggest that Perspective API has the potential for racial bias against speech performed by African Americans [110], but this possible bias should not compromise our analyses. A recent study systematically tested for adversarial examples and other recognized vulnerabilities in state-of-the-art multilingual models for toxicity detection and found that the latest production model of the Perspective API, the one used in this study, outperforms strong baselines [89]. Considering the widespread adoption of this tool and its state-of-the-art performance, we deem this tool suitable for use in this research.

All comments presented in Section Comments Datasets were pre-processed by the Perspective API, which assigns a continuous score between 0 and 1 to comments according to their toxicity. A higher score indicates a greater likelihood that a reader will perceive the comment as containing the given attribute, e.g., toxicity [77]. For example, as presented in the API documentation, a comment like "You are an idiot" may receive a probability score of 0.8 for the toxicity attribute, indicating that 8 out of 10 people would perceive that comment as toxic [77]. With this, it is possible to use this score, for example, to moderate toxic comments with a certain score [77]. Perspective architecture is composed of multilingual BERT-based models trained on data from online forums that are distilled into single-language Convolutional Neural Networks (CNNs) for each language that they support—distillation ensures the models can be served and produce scores within a reasonable amount of time [77]. Perspective has so-called production attributes, tested in various domains and trained on significant amounts of human-annotated comments. These attributes are available in English, Portuguese, and many

other languages. Also, it contains experimental attributes—English only—that are not recommended for professional use at this time [77]. In this work, we focus only on the toxicity attribute, the main feature among the production attributes. A comment with a high toxicity score is described in terms that resonate with the notion of incivility: "a rude, disrespectful or irrational comment that is likely to cause people to leave a discussion." To allow a fair comparison between datasets, toxicity scores were normalized using a min-max strategy for each dataset separately. Following a guideline presented in [78], we considered the identification of toxicity in comments in their original language. As noted above, while we believe that these measures of toxicity provide valuable proxies for incivility in the present study, future research may enrich this work by seeking to measure incivility directly.

S2 Appendix. presents a quick reference of summarized statistics of datasets and sources. The impact of cleaning noisy instances was greater for PARTISAN_pt sources. Despite this, the remaining dataset size is still large enough to consider in the analyses. Also, it is important to note that "extra.globo.com" source has the least amount of comments. Due to the sample's limited size, it has been excluded from subsequent analyses.

## Methods for hypothesis investigation

These are the three main methods used in this study:

- **Toxicity score distribution analysis:** We explore the distribution of toxicity scores on the different datasets and sources, separately. For that, we apply box plot visualizations.

- **Statistical test:** We conducted a statistical test to examine if there is a significant difference between toxicity scores across the evaluated datasets. Before defining an appropriate statistical test for this purpose, we check whether the data follows a normal distribution. To do this, we applied the D'Agostino K-squared test [111] and relied on Q-Q plots and histograms for each dataset to visually identify normal curve patterns (plots were not included for brevity). If data were non-normal, we apply the Kolmogorov-Smirnov Goodness of Fit test [112] pairwise between datasets. For this test, the hypotheses were $H_0$: datasets originate from the same distribution, and $H_1$: datasets do not originate from the same distribution.

- **Linear regression analysis:** We performed a linear regression analysis considering toxicity as a dependent variable and as the independent variable, the one we want to study in relation to toxicity.

Table 1 summarizes all datasets and methods used to study the hypotheses, while Table 2 cross-classifies the comparative datasets. Table 2 helps to organize the datasets and analysis by sorting them between media partisanship (neutral vs. partisan) and national polarization (high vs. low). The cells show datasets within each combination (e.g. highly partisan sources in a highly partisan context versus neutral sources in a less polarized context). As we were not able to clearly measure the combination of partisan sources with a less polarized national context due to the data limitations noted on Section Comparative Datasets (for H1, H2 and H3), we consider this cell to be an approximation (see also the note to Table 2).

## Results

In this section, we present the results regarding our hypotheses.

### Incivility and neutral bubble reachers

Fig 2 shows a box plot to help understand the difference in incivility (as measured by the toxicity scores) regarding the comparative and reference datasets. On this figure, it is possible to

**Table 1. Summary of all datasets and methods used to evaluate hypotheses.**

| Hypothesis | Datasets and Sources | Methods |
|---|---|---|
| **H1:** Neutral bubble reachers will tend to reduce uncivil discourse, relative to partisan accounts.<br><br>**H2:** Uncivil discourse will be more pronounced in contexts that are more populist and politically polarized.<br><br>**H3:** Neutral bubble reachers will produce greater uncivil discourse than partisan accounts in contexts that are populist and polarized. | **Comparative**<br>NEUTRAL_REACHER_pt<br>PARTISAN_pt<br>NEUTRAL_REACHER_en<br>PARTISAN_REACHER_en<br>**Reference**<br>CIVIL_pt<br>CIVIL_en<br>UNCIVIL_pt<br>UNCIVIL_en | **Toxicity distribution analysis** between datasets and respective sources, separately.<br>**Statistical test:** Kolmogorov-Smirnov Goodness of Fit test [98] pairwise between datasets. |
| **H4:** Uncivil discourse will be evident in both political and non-political topics in contexts that are populist and polarized. | **Alternative**<br>G1_SITE_pt<br>NYT_SITE_en | **Linear regression analysis** to verify the relationship between the source type (political or non-political) and toxicity scores. For this test, toxicity scores were considered as the dependent variable, and source type (political or non-political) the independent variable. |

note that the reference pairs UNCIVIL_pt and UNCIVIL_en are similar to each other, as well as the pairs CIVIL_pt and CIVIL_en, which indicates that these pairs share characteristics in common, despite linguistic and data source differences. As expected, the pairs UNCIVIL_pt and UNCIVIL_en proved to be uncivil, while CIVIL_pt and CIVIL_en proved to be the most civil, as also expected. These results indicate that the reference pairs present the desired characteristics for comparison with the other datasets. Furthermore, the slight difference observed within the reference pairs suggests that language might exert a minimal influence on the toxicity scores detected by the Perspective API in these extreme cases [78].

Fig 2 also highlights the difference between neutral bubble reachers and partisan accounts. The most surprising finding here is that, contrary to H1, NEUTRAL_REACHER_pt (neutral bubble reachers in Brazil) have more uncivil comments (according to their median toxicity scores) compared to PARTISAN_pt (partisan accounts in Brazil). By contrast, NEUTRAL_REACHER_en (neutral bubble reachers in Canada) fits the expectations of H1: the neutral bubble reachers inspire less uncivil comments than their partisan counterparts in

**Table 2. Cross-reference of comparative datasets applied for H1, H2 and H3 hypotheses testing.**

| | | National polarization | |
|---|---|---|---|
| | | **High** | **Low** |
| **Media type** | **Neutral Bubble Reacher** | **NEUTRAL_REACHER_pt** ($N = 123, 212$) (**Brazil**):<br>• extra.globo.com ($N = 48$—removed from analysis due it's size)<br>• g1.globo.com ($N = 55, 057$)<br>• noticias.uol.com.br ($N = 68, 155$) | **NEUTRAL_REACHER_en** ($N = 113, 114$) (**Canada**):<br>• globalnews.ca ($N = 2, 038$)<br>• cbc.ca ($N = 97, 932$)<br>• theglobeandmail.com ($N = 13, 144$) |
| | **Partisan** | **PARTISAN_pt** ($N = 24, 741$) (**Brazil**):<br>• CampanhadoArmamento ($N = 4, 199$)<br>• MisesBrasil ($N = 3, 508$)<br>• OPesadelodeQualquerPolitico2.0 ($N = 4, 306$)<br>• Carvalho.olavo ($N = 4, 218$)<br>• Flaviobolsonaro ($N = 4, 307$)<br>• padrepaulo ($N = 4, 203$) | ——— **Approximation** ———<br>**PARTISAN_REACHER_en** * ($N = 28, 070$) (**Canada-USA**):<br>• huffington_post ($N = 13, 013$)<br>• ny_times ($N = 14, 169$)<br>• bloomberg_politics ($N = 888$) |

*This dataset was obtained by cross-referencing the top 100 biased bubble reachers in Canada (measured by their $RP(H)$) with a Facebook dataset containing comments on 20, 000 posts made by American news organizations and pers onalities [99]. See Comparative Datasets (for H1, H2 and H3) section for a complete reference. Note that this dataset represents an approximation for the Canadian scenario.

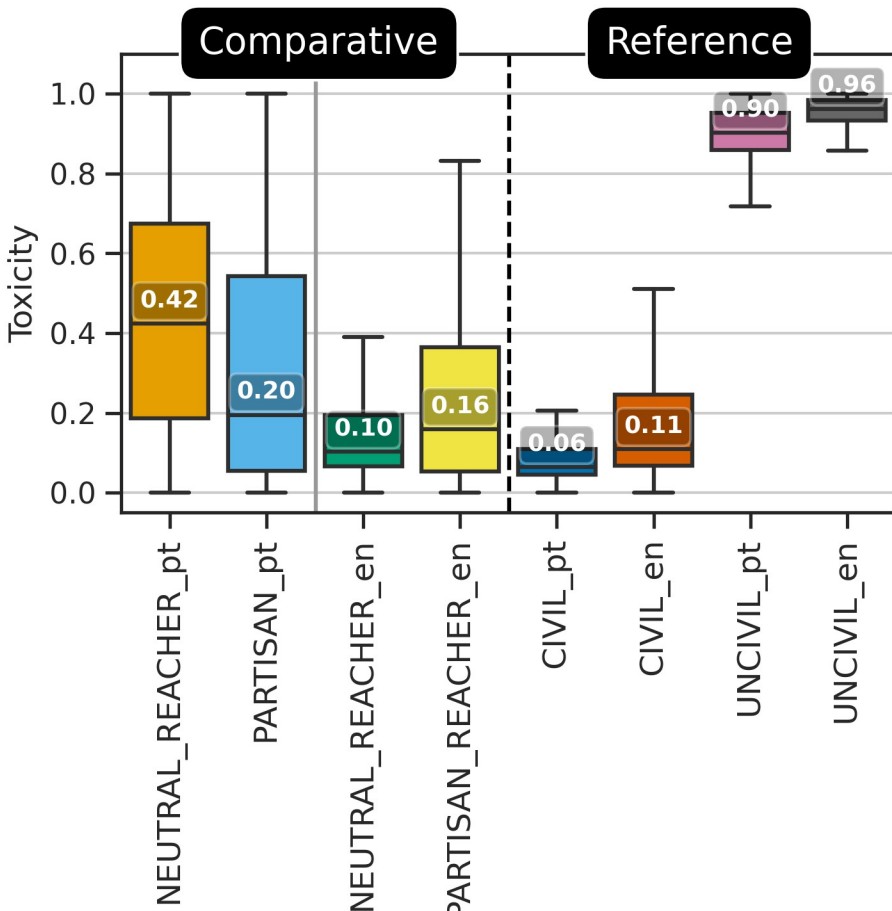

**Fig 2. Box plots showing the distribution of incivility (measured by toxicity scores) for the comparative and reference datasets.**

PARTISAN_REACHER_en (an approximation for partisan accounts related to Canada). This contrast between the more polarized Brazilian context and the less polarized Canadian one suggests support for H3: neutral bubble reachers in the polarized context generate more uncivil discourse than partisan sources, whereas in the Canadian context, the opposite is the case (regarding this result, it is important to recall that PARTISAN_pt is mostly composed by comments made on content published by extreme right Facebook pages which probably do not promote an encounter between distinct political views like what could be happening on NEUTRAL_REACHER_pt, which is composed mostly by comments on news accessed by users on distinct positions on the political spectrum). Just as intriguing is that the median toxicity score on the Canadian dataset (NEUTRAL_REACHER_en) is close to the median for CIVIL_en and CIVIL_pt references, while all these three datasets have, at least, 3.8 times lower toxicity median score compared to NEUTRAL_REACHER_pt, and, at least, 1.8 times lower toxicity median score compared to PARTISAN_pt. These results lend support to H2: uncivil discourse, in general, is lower in the less polarized and populist Canadian context relative to the more polarized Brazilian setting.

To verify whether differences between median toxicity scores observed in Fig 2 were statistically significant, we applied the tests explained in Section Methods for Hypothesis

**Table 3. D'Agostino K-squared ($k^2$) tests for comparative datasets.**

| Dataset | $k^2$ |
|---|---|
| NEUTRAL_REACHER_pt | 471793.46*** |
| NEUTRAL_REACHER_en | 45481.83*** |
| PARTISAN_pt | 4763.45*** |
| PARTISAN_REACHER_en | 4285.82*** |

Note: $k^2$ represent the D'Agostino K-squared [111] statistic. All statistics are significant at $p < .001$ (***) level.

Investigation. First, Comparative datasets (NEUTRAL_REACHER_pt, NEUTRAL_REACHER_en, PARTISAN_pt, and PARTISAN_REACHER_en) were subjected to a normal test for the toxicity scores applying D'Agostino K-squared test [111]. Table 3 presents these test results, showing that none of the datasets followed a normal distribution ($p < .001$). We also relied on Q-Q plot and histogram visualizations for each dataset, confirming this result (plots were not included for brevity). Considering that the data were not following a normal distribution, we applied the Kolmogorov-Smirnov Goodness of Fit test [112] pairwise between datasets to verify whether the samples originate from the same distribution. Results for this test were included in a separate spreadsheet for consultation (https://zenodo.org/records/10443022), which shows that none of the comparative datasets originated from the same distribution ($p < .05$), meaning that the observed differences on box plots are statistically significant.

These results suggest that, in general, there does not appear to be a simple and direct relationship between being a neutral bubble reacher and reducing uncivil discourse across all situations. Rather, the relationship between "reaching the bubble" and incivility appears to depend on the broader national context in which neutral and partisan accounts operate. Table 4 summarizes these results along the same lines as Table 2, organizing them by media source partisanship and national polarization.

We conducted a number of additional analyses to increase our confidence in these results. These are reported in S4 and S5 Appendices. It is possible that specific sources influence the results. Therefore, in S4 Appendix we decompose the analysis by all individual sources. We found that only in the Canadian context were there more prominent differences across sources, with globalnews.ca showing a higher toxicity score than other Canadian sources. Possible reasons for that difference include the fact that, relative to globalnews.ca, the other organizations in the Canadian dataset are more traditional and established sources, whereas Global News emerged more recently and grew out of talk radio. It is also possible that the difference arises from the fact that globalnews.ca integrates its discussion board automatically with Facebook, and Facebook comments could tend to be less civil than those on the internal news

**Table 4. Summary of results for H1, H2 and H3.** Plus (+) sign indicates the incivility level, ranging from + + + + + (highest) to + (lowest).

| | | National polarization | |
|---|---|---|---|
| | | **High** | **Low** |
| **Media type** | **Neutral Bubble Reacher** | **HIGHEST INCIVILITY** (+++++)<br>Neutral bubble reachers elicited the highest incivility compared to all other cases in a highly polarized situation (Brazil). | **LOWEST INCIVILITY** (+)<br>Neutral bubble teachers elicited the lowest incivility in a less polarized situation (Canada). |
| | **Partisan** | **MODERATE TO HIGH INCIVILITY** (++++)<br>Partisans also elicited considerable incivility, but slightly lower when compared to neutral bubble reachers in a highly polarized situation (Brazil). | **LOW TO MODERATE INCIVILITY** (+++)<br>Partisans elicited some incivility, which was slightly higher than bubble reachers in a less polarized situation (Canada-USA). |

organization systems, which are often moderated and require users to register. For this reason, we evaluate this possibility in S5 Appendix and find little evidence that uncivil comments are differentially related to Facebook than elsewhere, giving us greater confidence in our results. This analysis also indicates that in the case of globalnews.ca, other factors could be responsible for greater incivility. This would be a topic worth pursuing in an analysis more specifically focused on the Canadian media ecosystem.

## The politicization of non-political content in a polarized context

The preceding analysis suggests ideologically neutral bubble reachers operate differently in Canada and Brazil. One possibility is that the heightened polarization in the Brazilian context makes any performance of ideological neutrality more suspect, resulting in political contention seeping into even seemingly non-political discussions. H4 explores this possibility by examining how incivility levels vary by political vs non-political content shared by ideologically neutral bubble reachers. Scholars find that political content elicits more uncivil discourse. However, evidence of minimal toxicity score discrepancy between political and non-political content would suggest this distinction is blurred in a highly polarized context. If polarization closes the room for neutrality, this could explain why ideologically neutral bubble reachers are unable to temper uncivil discourse in Brazil.

To examine H4, we investigated comments on the G1_SITE_pt, collected from the portal at g1.globo.com. This dataset is composed of comments extracted from the same website as g1.globo.com source included in the NEUTRAL_REACHER_pt dataset, but at distinct moments. This analysis was conducted to identify whether there are differences between comments made about politics in a polarized situation, represented by the g1.globo.com source from NEUTRAL_REACHER_pt dataset, and comments made about distinct topics and situations, represented by the G1_SITE_pt dataset and underlying sources (each of which representing news columns on G1_SITE_pt). Fig 3 shows the distribution of toxicity scores for the G1 sources.

These box plots present some initially surprising results. Consistent with previous literature, political content generates high toxicity scores. G1_SITE_pt "Política"—Political, translated to English—is among the most uncivil columns in G1_SITE_pt. However, in line with H4, the differences in its toxicity scores compared to some of the other subjects are minimal. Consider "Bem estar" (Wellness), "Educação" (Education), and "Agro" (Agriculture), translated to English. These columns show a high toxicity score, which is surprising, given that they appear to contain relatively neutral topics. To try to understand these cases, we analyzed samples of comments with the highest toxicity score in each of these columns. Our findings revealed that these instances predominantly involved discussions on the political landscape surrounding health, education, or agriculture. Thus, this suggests that H4 holds, that even apparently neutral subjects are assimilated into political conflict in a highly polarized context (see [113] for a recent study finding that Political posts from influential users on Twitter engage users more than non-political posts).

To verify if the differences between toxicity scores observed in Fig 3 were statistically significant, we applied the tests explained in Section Methods for Hypothesis Investigation—the same one applied for datasets in Section Incivility and neutral bubble reachers, but changing the tested variables to be the G1 sources toxicity scores presented in this figure. These G1 sources were subjected to a normal test applying D'Agostino K-squared test [111]. Table 5 presents these test results, showing that none of the Facebook sources followed a normal distribution ($p < .001$). We also relied on Q-Q plots and histogram visualizations for each G1 source, confirming this result (plots were not included for brevity). Considering that the data

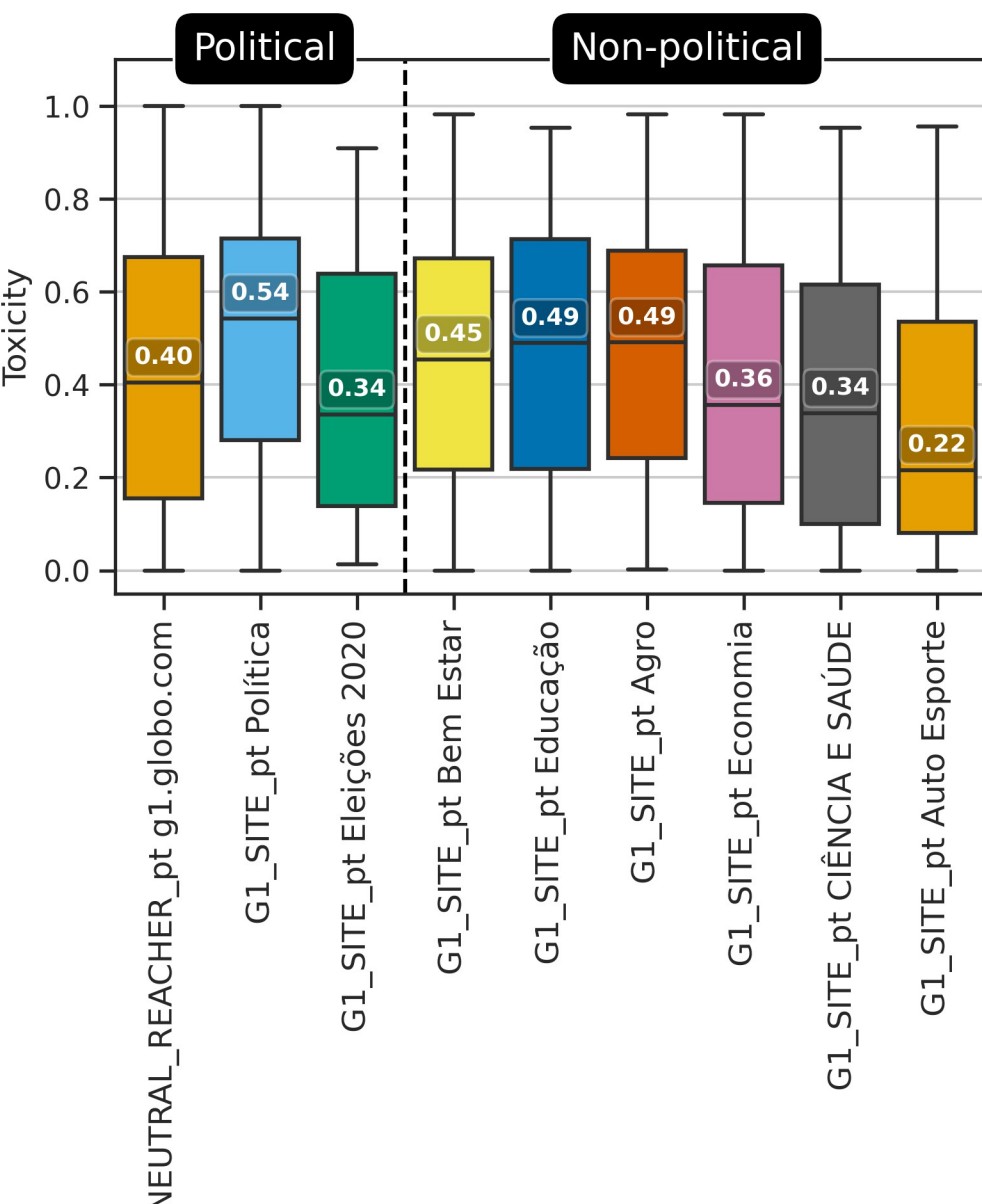

**Fig 3. Box plots showing the distribution of toxicity scores for the G1 sources.**

were not following a normal distribution, we applied the Kolmogorov-Smirnov Goodness of Fit test [112] pairwise between G1 sources to verify whether the samples originate from the same distribution. Results for this test were included in a separate spreadsheet for consultation (https://zenodo.org/records/10443022), which shows that none of the G1 sources originated from the same distribution ($p < .05$), except for two cases: the pairs "Eleições 2020" and "Economia", as well as, "Eleições 2020" and "CIÊNCIA E SAÚDE". These cases, as can be seen in the box plots in Fig 3 have similar distributions and medians, so this was expected and does not impact the conclusion. Therefore, the observed differences in box plots are statistically significant, except for these specific cases.

**Table 5. D'Agostino K-squared ($k^2$) tests for G1 sources.**

| Dataset | Source | $k^2$ |
|---|---|---|
| NEUTRAL_REACHER_pt | g1.globo.com | 982481.041*** |
| G1_SITE_pt | Política | 235072.0699*** |
| | Eleições 2020 | 2163.3942*** |
| | Bem Estar | 137482.9471*** |
| | Educação | 118091.6763*** |
| | Agro | 2445.8281*** |
| | Economia | 115085.351*** |
| | CIÊNCIA E SAÚDE | 3860.0542*** |
| | Auto Esporte | 252.9476*** |

Note: $k^2$ represent the D'Agostino K-squared [111] statistic. All statistics are significant at $p < .001$ (***) level.

To study H4 more formally, we examined the influence of political topics on the toxicity score of comments with regression analyses regarding political and non-political comments in the G1_SITE_pt and NYT_SITE_en datasets (see Section Methods for Hypothesis Investigation). Like Brazil, the American context from which NYT_SITE_en was collected is regarded as more polarized. In both cases, the linear regression coefficients were statistically significant ($p < .001$), being $r = 0.116$*** for G1 and $r = 0.093$*** for New York Times. Still, while statistically significant, the influence was weak for both cases, showing that political topics' influence on incivility is minimal. Overall, this result indicates that political topics seem to elicit slightly more uncivil comments compared to non-political topics.

## Discussion and conclusion

Bubble reachers manage to reach ideologically diverse users by distributing information across politically distinct homophilic groups, but those who are exposed to such content tend to share what is aligned with their own political orientation [20]. The present study extended these findings, moving beyond the political valence of content that users share to the discursive style of that content. In particular, we examined a series of hypotheses that stem from asking whether ideologically neutral bubble reaching accounts create venues where users engage in more civil discourse, as compared to partisan accounts.

Our results complicated the simple hypothesis of a universal impact of neutral bubble reachers across contexts. Instead, we found evidence that the relationship between bubble reaching and incivility is moderated by the national political culture. In the Canadian case, we find that ideologically neutral bubble reachers tend to elicit more civil discourse. Compared to more ideologically partisan outlets, neutral bubble reachers carve out discursive spaces where uncivil commentary is less pronounced. This is consistent with the stated goals of these outlets, where aspirations towards objectivity and fairness serve as a foundation for more civil discourse. However, incivility was more pronounced in the Brazilian context among the ideologically neutral bubble reachers.

By exploring further hypotheses, we found evidence as to the source of these differences. It appears that the capacity of neutral bubble reachers to operate as mediating institutions may be undermined in highly-polarized contexts where populist discourse is prominent. Bubble reachers may present themselves as neutral arbiters of impartial expertise, but this performance may backfire under these conditions [57]. As discussed in the literature review, populists often undermine trust in mediating institutions by arguing those institutions and the

elites who staff them advance the particular interests of the status quo. Polarization, more generally, raises the stakes of politics, creating a climate where passionate partisanship is incentivized while remaining neutral or unbiased becomes treated with suspicion. This context works against the performances of ideologically neutral bubble reachers, constraining the discursive space for civility.

The greater prominence of populism and polarization in the Brazilian context may explain the discrepancies we observe. Consistent with this depiction, our analysis finds greater levels of incivility in the Brazilian context across both neutral and partisan accounts, suggesting a more inflamed discursive space in general. We also find that while explicitly political content tends to elicit somewhat more uncivil discourse, the discrepancy is lower than expected: even ostensibly non-political content gets pulled into the ambit of partisan politics. Collectively, this may explain the role of ideologically neutral bubble reachers in Brazil—populism and polarization challenge these outlets' credibility, turning discursive spaces into meeting points for rancorous conflict rather than mediation, and shrinking the room for civility. Under these circumstances, ideologically neutral bubble reachers may "open the hand" for broader discussion, but still receive the fist.

Our findings support and extend research on the contextual nature of incivility. Scholars argue that incivility is inherently sensitive to cultural context—prevailing interaction norms dictate whether an expression is perceived to be disrespectful or not [114]. But even those who breach norms of civility do so in ways that are attuned to contextual factors at different levels [115]. While several studies point to variation in topical context of news stories as an important factor [24, 63], the role of national context remains unclear. One of the few comparative studies, in fact, finds that while levels of uncivil conflict vary by national context, the relationship between incivility and political engagement is not mediated by national context [116]. In contrast, we find that national context mediates where incivility is more likely to be expressed —whether in relation to neutral bubble reaching outlets, or across topical issues. While Otto et al.'s [116] comparative analysis focused on three European countries, our findings suggest that when examining countries with more dissimilar political cultures, the patterning of incivility may differ. Further systematic comparative analysis exploring how incivility is mediated in different national contexts would be a worthwhile direction for scholars to pursue.

That the neutral bubble reachers detected by our analysis corresponded with legacy media outlets also makes our findings relevant to related scholarship. Research suggests that hostile media perceptions—the tendency of partisans to view media coverage as biased against them even if coverage is even-handed—contribute to incivility in discourse [58, 117]. Scholars highlight cognitive, emotional, and behavioural dimensions of this process: motivated reasoning distorts how partisans receive information from outlets, contributing to perceptions of bias that incite feelings of media indignation, and increase willingness to engage in uncivil discourse in turn [118]. Our findings are consistent with this work, showing how greater incivility can ensue as legacy media outlets become meeting-points for partisan conflict. We extend this work, however, by underscoring the importance of cultural context in cueing this package of cognitive, emotional, and behavioural processes. The underlying mechanisms are not entirely clear. For instance, a populist context may frame the reputation of legacy media outlets in ways that disproportionately attract antagonistic partisans while alienating moderates, heightening incivility as a result of altering the composition of commenters [119]. Alternatively, cultural norms in a populist context may heighten media indignation towards these outlets across the board, increasing the expression of incivility for both partisans and moderates. Future research can work towards clarifying the mechanisms whereby cultural context mediates the relation between legacy media and incivility.

This study represents a significant advancement from previous research by analyzing the communication style provoked by neutral bubble reachers. The findings from this study could help researchers, platform designers, and policymakers seeking to promote healthier online interactions. Recognizing the contextual factors that impact the effectiveness of neutral bubble reachers can inform the development of strategies to mitigate incivility and foster more constructive dialogue in online spaces. Moreover, understanding the interplay between polarization, populism, and the performance of mediating institutions can contribute to efforts to address the broader challenges associated with political polarization and the erosion of civility in public discourse.

While our study is highly suggestive of these relationships, some possible limitations should be highlighted. One was regarding the PARTISAN_REACHER_en dataset, which was used to approximate Canada's less polarized political culture. This approximation was achieved by cross-referencing the top 100 bubble reachers with partisan $RP(H)$ on Canada with a Facebook dataset containing comments on 20, 000 posts made by American news organizations and personalities [99]—see Comparative Datasets (for H1, H2 and H3) section for a complete reference.

The fact that the comparative, reference and alternative datasets were obtained from different sources makes comparisons less robust, since the dynamics of interactions on each platform can influence the degree of incivility. For example, NEUTRAL_REACHER_pt and NEUTRAL_REACHER_en comments were extracted from content produced on distinct news websites, while PARTISAN_pt and PARTISAN_REACHER_en comments were extracted from content on Facebook Pages in distinct situations and dates. In this sense, each source (news website) has distinct moderation mechanisms that may or may not allow uncivil behaviour in comments, thus making some news outlets less uncivil by default. These differences make it difficult to conduct a perfectly fair comparison between them. In this same direction, cultural, contextual and linguistic differences may also directly affect the toxicity we observe in comparing datasets in distinct languages and situations. Additionally, it is noteworthy that PARTISAN_pt exclusively comprises right-wing pages, whereas PARTISAN_REACHER_en includes left-wing or left-leaning pages, as indicated by the All Sides Media Bias Report. This distinction is pertinent, as varying political perspectives may prompt disparate online behaviours in terms of information consumption [108, 120] (e.g. liberals being more likely than conservatives to engage in cross-ideological dissemination) [41]). Considering these aspects, the differences that can be observed in datasets and sources and conclusions that were drawn from these results need to be considered with caution. We tried to address some of these limitations by including additional analyses in S5 and S6 Appendices.

Regarding differences between datasets and sources, it is worth noting that the standard deviation for some datasets, such as, for example, NEUTRAL_REACHER_pt, PARTISAN_REACHER_en and PARTISAN_pt datasets is high (S2 Appendix). This can potentially limit conclusions about these cases, even when there are statistically significant differences between them. We also recognize that the pre-processing step, although not aggressive, may have eliminated some representative comments in some datasets, such as PARTISAN_pt. Despite this, the number of comments in our analysis was high in all datasets and sources, which reduces the chance that the pre-processing step greatly influenced the results. Another limitation refers to the polarization characteristics in the NEUTRAL_REACHER_pt and NEUTRAL_REACHER_en datasets. In this sense, several comments presented subtle uncivil characteristics, mainly with sarcasm related to the political context itself. These subtleties could be too complex to be captured by a generic model, perhaps even by humans without detailed local context-sensitivity [78].

Our study opens up several possibilities for future work. For instance, while we analyzed political polarization in two different contexts, it is essential to investigate more contexts, including similar contexts in other countries and new contexts. Extending our analysis beyond Canada and Brazil could provide more robust support for the observed role of political culture in mediating how bubble reachers operate. Further, examining the hypotheses in different periods for the same country is also an important future step; this enables the understanding of, for example, how changes in political culture affect the results. For instance, we explain higher incivility levels in Brazil due to greater polarization and populism in their political culture—evidence of longitudinal growth in incivility corresponding with polarization would be in line with this explanation and could parse out the time-ordering between these two. Likewise, a longitudinal analysis could examine how the diffusion of populist framing in public culture affects whether citizens adopt more toxic styles in their expression [121]. Another motivation for examining our hypothesis in a different period is because COVID-19 has been associated with an increase in affective polarization, exacerbated by factors such as the politicization of public health measures and information, leading to heightened ideological divides [122]. Although most of our data do not cover this period (only G1_SITE_pt and possibly UNCIVIL_pt, which does not provide their collection date), polarization could have changed after this period. Additionally, given that the neutral bubble reacher accounts examined were comprised of legacy news outlets, it is essential to ponder whether similar conclusions can be extended to other account types, such as bubble reaching accounts entirely unrelated to politics. These expanded possibilities contribute to the generalizability of the findings across diverse political cultures and various types of bubble reaching accounts.

## Supporting information

**S1 Appendix. Alternative datasets used in sensitivity analysis [123].**
(PDF)

**S2 Appendix. Datasets and associated sources summary [20, 99, 102–105, 123].**
(PDF)

**S3 Appendix. Comments pre-preprocessing [124–126].**
(PDF)

**S4 Appendix. Sensitivity analysis—Toxicity by sources [111, 112].**
(PDF)

**S5 Appendix. Sensitivity analysis—Incivility on Facebook by different accounts [111, 112].**
(PDF)

**S6 Appendix. Sensitivity analysis—Facebook influence on incivility.**
(PDF)

## Acknowledgments

We want to thank the authors who made their data available, enabling some analysis of this study. We also want to thank all the anonymous reviewers for their valuable feedback.

## Author Contributions

**Conceptualization:** Jordan K. Kobellarz, Milos Brocic, Daniel Silver, Thiago H. Silva.

**Data curation:** Jordan K. Kobellarz.

**Formal analysis:** Daniel Silver.

**Funding acquisition:** Thiago H. Silva.

**Investigation:** Jordan K. Kobellarz, Milos Brocic, Daniel Silver, Thiago H. Silva.

**Methodology:** Jordan K. Kobellarz, Milos Brocic, Daniel Silver, Thiago H. Silva.

**Project administration:** Daniel Silver, Thiago H. Silva.

**Software:** Jordan K. Kobellarz.

**Supervision:** Daniel Silver, Thiago H. Silva.

**Validation:** Jordan K. Kobellarz.

**Visualization:** Jordan K. Kobellarz.

**Writing – original draft:** Jordan K. Kobellarz, Milos Brocic, Daniel Silver, Thiago H. Silva.

**Writing – review & editing:** Milos Brocic, Daniel Silver, Thiago H. Silva.

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
