## [Decision Letter · Decision Letter 0]

24 May 2023

PONE-D-23-09232Bubble reachers and toxic discourse in polarized online public spherePLOS ONE

Dear Dr. Silva,

Thank you for submitting your manuscript to PLOS ONE. After careful consideration, we feel that it has merit but does not fully meet PLOS ONE’s publication criteria as it currently stands. Therefore, we invite you to submit a revised version of the manuscript that addresses the points raised during the review process.

We look forward to receiving your revised manuscript.

Kind regards,

Michal Ptaszynski, PhD

Academic Editor

PLOS ONE

Journal Requirements:

“All stages of this study were financed in part by the project SocialNet (process 2023/00148-0 from S˜ao Paulo Research Foundation - FAPESP), and CNPq (grant 310998/2020-4)”

“TS and JK - project SocialNet (process 2023/00148-0 from Sao Paulo Research Foundation - FAPESP)

TS - CNPq (grant 310998/2020-4).

Reviewers' comments:

Reviewer's Responses to Questions

**Comments to the Author**

1. Is the manuscript technically sound, and do the data support the conclusions?

Reviewer #1: Yes

Reviewer #2: No

2. Has the statistical analysis been performed appropriately and rigorously? 

Reviewer #1: Yes

Reviewer #2: Yes

3. Have the authors made all data underlying the findings in their manuscript fully available?

Reviewer #1: Yes

Reviewer #2: Yes

4. Is the manuscript presented in an intelligible fashion and written in standard English?

Reviewer #1: Yes

Reviewer #2: Yes

5. Review Comments to the Author

Reviewer #1: REVEW: Bubble Reachers and Toxic Discourse in Polarized Online Public Sphere.

Submitted to PLOS One

Manuscript #: PONE-D-23-09232

This study examines the impact of “bubble reachers on the civility/toxicity of online discussions in two distinct contexts: Brazil (marked by high levels of populism) and Canada (conversely marked by low levels of populist sentiment). The study tests a series of insightful hypotheses and concludes that the impact of bubble reachers is largely situation (i.e. “moderated by the national political context”). Specifically, network brokers appear to contribute to less toxic discourse in contexts that lack strong partisan sentiments but invite more toxic discourse in contexts with greater populist sentiment.

On the whole, I find the study to be well-done and compelling, and I believe that it will make a valuable contribution to the literature. I do think that there are some revisions/considerations that should be undertaken before finalizing the manuscript for publication. These are outlined below:

1. There are two issues surrounding language and framing that I would recommend the authors reconsider:

a. The outcome variable is framed largely in terms of “toxic discourse”, but I’m not persuaded that this is the most appropriate construct for approaching the questions at hand. In it’s simplest terms, the issue at stake here is whether politically neutral brokers/bridges in a network lead to more civil, democratic deliberations or whether they contribute to incivility and hostile interpretations of content/information. It seems that “civility” and “democratic discourse” are more commonly accepted and clearly defined outcomes. It’s not necessarily self-evident that “toxicity” is the antithesis of deliberation, and it’s been argued in some contexts that effective democracy can be “rude”. At a minimum, I would recommend that the authors be a bit more deliberate (though this can be brief) about connecting these concepts and addressing the value of democratic deliberation (regardless of its tone). Ideally, it would be more compelling if the findings could be discussed in the context of deliberation and its import.

b. The other area that I would recommend considering is the language around “bubble reachers”. This may be an artifact of different disciplinary groundings, but in network studies the more traditional terms would be “brokerage/brokers” and/or “network bridges”. This may create some confusion for diverse disciplinary audiences. At a minimum, I would recommend connecting/explaining this diverse terminology early in the manuscript. (There are some areas where the authors use the term “Brokerage”, but I’m not sure the terminology is ever clearly explained in an inter-disciplinary manner).

2. There are also a few additional areas where I would recommend briefly building out the literature in order better situate the findings across diverse disciplinary perspectives. I think that the authors’ findings will resonate deeply with those who study confirmation bias and hostile media effects. I would recommend leaning a bit more into the language of both literature bases (both to better contextualize the findings and to increase the study’s exposure across niche disciplines). There is also a bit more literature to consider re: the “echo chamber hypothesis” (such as recent work by Dubois and Blank, 2018) which suggests that the societal impacts of online filter bubbles may be overstated (though I would argue that there is still reason to be concerned about this phenomenon).

3. There are some areas (particularly early in the manuscript) that need additional citations. Some examples include “Plato” (pg. 2), “These, to some scholars, evoked the forms of” (pg. 2), “A common response… has been to encourage exposure…” (pg. 2); “the claim to objectivity… provoke greater scorn as a result” (pg. 7), etc.

4. The authors note that data include comments related to “articles published during the electoral period in each country”. This period should be clearly defined for each country, as should the context of the elections in question. The authors do an excellent job of framing each country’s context as it relates to populist sentiment, but what about the specific context of these elections? Were they tightly contested elections? Were they “landslide” elections? Were there specific economic or social conditions that drove electoral debates? And do these factors affect the interpretation of findings in any way?

5. Lastly, I did find the naming of data sets to be a bit hard to follow. The data sets are similarly named and I often found myself having to go back and forth to remind myself which was which. I wonder if the use of more conceptual terminology when referring to the data sets would be easier for readers. (i.e. Neutral Brazilian Sources, Partisan Brazilian Sources, etc.)

a. I would also recommend including a little more specific info about who these sources are (perhaps in the “Supporting Information”).

Again, I think that the study makes a valuable contribution to the literature. My comments and recommendations are largely focused on clarifying some terminology, improving the readability of the manuscript, and better situating the findings across relevant disciplines.

Reviewer #2: This paper aims to compare levels of toxicity in comments using two case-studies, Canada and Brazil, to advance the idea that some media outlets and accounts may serve as ‘bubble reachers’ and breach into polarized networks. While the focus on a comparative case is welcome and so is the inclusion of the Brazilian case, as there is still little research focused on global south countries, the contributions of this specific project are not clear to me for both conceptual and methodological reasons.

Conceptually, the idea of bubble reachers and the operationalization of these accounts seem to come from another study by some of the authors, and the current paper simply takes on accounts identified as bubble reachers and conducts an analysis of their comments, doing little to expand on, or demonstrate, the role of bubble reachers. Second, the study that drives the conceptualization and operationalization of bubble reacher accounts is based on Twitter data, not Facebook, and network features and associations across the two platforms are sufficiently different to warrant a validation of the concept, its ‘representative accounts’, and its role in a different network. That this may not be possible due to data collection restrictions on Facebook is understandable—however, in that case, what can we really say about bubble reachers on Facebook and what is the validity of extending the observations made on Twitter to this platform considering such limitations? Another conceptual issue, that spans into the methodology, is the adoption of an ‘off-the-shelf’ classifier for toxic discourse that is disconnected from broader discussions about online incivility and implies that ‘toxic’ = bad, perhaps leaning on an outdated view of the ‘online public sphere’ that is driven by a scarce literature review. Still in the realm of conceptual and framing issues, the paper often refers to populist discourse to explain the results in Brazil, but that perspective is not grounded in any data and seems to suggest that the problem in the country is right-wing populism, whereas populism, as typically measured in surveys, is typically quite high in Brazil regardless of political leaning, being a weak explanatory concept to understand the underlying dynamics that the paper seems to unveil.

In terms of the methodology, (1) there is no clear rationale as to why you have datasets from different platforms that are no comparable across categories, e.g. with biased datasets coming from Facebook pages and the ‘neutral’ datasets coming from comments in pages. Moderation practices across platforms are quite different because even news outlets are limited in the ways they are able to moderate content on Facebook, whereas they typically employ heavier moderation strategies on their own websites. In the same vein, one explanation for some results in Canada has to do with a website using Facebook for comments, which is not comparable to native comments. The methods section seems to ignore such discrepancies and does not provide a convincing explanation for the approach. (2) and related to the point about toxicity above, using off-the-shelf algorithms to classify text at scale is not sufficient to contribute to the understanding of online toxicity as a societal issue, as it does not advance our knowledge about context or role of toxic discourse online. Many have argued that online incivility may relate to ‘deliberative’ discourse traits, and research on perceptions of incivility demonstrates that individuals have a wide variation in the types of expressions that they see as uncivil. As such, a single measure of toxicity based on an algorithm that has not even been validated in the data it’s been applied to is not particularly enlightening to learn about the dynamics underlying polarization and public discourse. Importantly, polarization here seems to be related to a country characteristic: there are no measures of polarization and the biased datasets appear to be one-sided, mainly including right-wing accounts.

All in all, I struggle to see a scientific advancement of our collective understanding of a) online toxicity and b) its relationship with polarization based on an analysis of an off-the-shelf classifier on a subset of accounts that supposed to be ‘neutral’ or biased based on other studies in other platforms. This study does not advance the understanding of news accounts and comments, nor toxicity classification, nor actual political polarization.

6. PLOS authors have the option to publish the peer review history of their article (what does this mean?). If published, this will include your full peer review and any attached files.

Reviewer #1: No

Reviewer #2: No

---

## [Author Response · Author response to Decision Letter 0]

12 Jul 2023

***Please note that we submitted a pdf file with the responses*****

---

## [Decision Letter · Decision Letter 1]

9 Oct 2023

PONE-D-23-09232R1Bubble reachers and toxic discourse in polarized online public spherePLOS ONE

Dear Dr. Silva,

Thank you for submitting your manuscript to PLOS ONE. After careful consideration, we feel that it has merit but does not fully meet PLOS ONE’s publication criteria as it currently stands. Therefore, we invite you to submit a revised version of the manuscript that addresses the points raised during the review process.

We look forward to receiving your revised manuscript.

Kind regards,

Michal Ptaszynski, PhD

Academic Editor

PLOS ONE

Reviewers' comments:

Reviewer's Responses to Questions

**Comments to the Author**

1. If the authors have adequately addressed your comments raised in a previous round of review and you feel that this manuscript is now acceptable for publication, you may indicate that here to bypass the “Comments to the Author” section, enter your conflict of interest statement in the “Confidential to Editor” section, and submit your "Accept" recommendation.

Reviewer #1: All comments have been addressed

Reviewer #3: (No Response)

2. Is the manuscript technically sound, and do the data support the conclusions?

Reviewer #1: Yes

Reviewer #3: Partly

3. Has the statistical analysis been performed appropriately and rigorously? 

Reviewer #1: Yes

Reviewer #3: No

4. Have the authors made all data underlying the findings in their manuscript fully available?

Reviewer #1: Yes

Reviewer #3: Yes

5. Is the manuscript presented in an intelligible fashion and written in standard English?

Reviewer #1: Yes

Reviewer #3: No

6. Review Comments to the Author

Reviewer #1: I appreciate the opportunity to re-review the manuscript, and I appreciate the rigor and effort that the authors have put into the revisions. Basing my response on the initial comments that I provided, I believe that the authors have effectively addressed my concerns (as well as those raised by the other reviewers).

The only area where I would still encourage some minor revisions is the issue of better connecting these findings to the literature around confirmation bias and hostile media effects. The authors made some minor notes on this, but the connections remain a bit under-developed. I still believe that there is a rich opportunity here to better connect to this important area/body of literature. I would encourage the authors to consider an additional paragraph in the discussion addressing this. Can we make any inferences from these findings that might apply to the literature around confirmation bias? Does this study point to any novel opportunities for future research in that area?

Beyond that, I believe that my prior comments have been well-addressed.

Reviewer #3: Review for Bubble reachers and toxic discourse in polarized online public sphere

The authors present a complex study with several distinct data sets aimed at testing the relationship between polarization on the national level and toxicity in the comment sections of two different types of communicators: large scale rather neutral news outlets as compared to more partisan outlets. Albeit I do think that there is clearly some merit in the paper, there are several aspects that prevent me from recommending its publication in the current state. I know that this is surely not the result the authors had hoped for, moreover as this a revised manuscript and I am new on the board. I will thus refrain from reiterating the comments made by the other reviewers before. They are all legit and I am not sure if they were addressed all sufficiently, but it is an editorial task to judge this. Instead, I will provide some additional nuances (albeit they do reiterate the sentiment from the already existing reviews). I hope that they will find my comments useful in polishing their work.

(1) Theoretical rigor. Albeit I do understand the disciplinary origins of the term bubble reacher, I do not think that it does a good job in linking the study to the existing research out there and I am not convinced of the implications of the metaphor. I have two reasons for my concern: (a) the “neutral bubble reachers” seem to be basically legacy news sources. So why not call them like that? This would allow tons of social sciences to relate to the findings as they potentially tell us larger stories about public spheres (something communication science is interested in), the normative ideal of deliberation vs. conflicts (something political philosophy will be interested in), or the role of news in polarized contexts (which would be of interest to political science amongst others).

Through the analogy of the bubble reacher, the (pretty much debunked) concept of the filter bubble is reiterated despite exactly these accounts contradicting the original filter bubble hypotheses. And there is no evidence that fragmentation itself does foster toxic discourse (p.5). This strongly depends on the communication norms in the disparate communities (Gibson, 2019).

Also, readers will find it likely easier to understand what legacy news vs. partisan news are, while bubble reacher is a new term they need to recall over and over again. Furthermore, to convince me of the term, I would like to see the generalizability of findings to other “bubble reachers” out there such as completely unpolitical accounts. Do the hypotheses hold for Taylor Swift too? This question seems to be simple, but the authors provide the (interesting) idea that “[bubble reachers] offer spaces that mitigate rancorous discourse by providing a common starting point for conversation, even if that starting point is differently interpreted? Or do they provide the discursive equivalent of “combat zones,” gladiatorial arenas where partisans meet not to deliberate and converse, but to fight?” – This is basically the idea of either deliberative or confrontative theories about the public sphere (Habermas, 2006; Mouffe, 2018). Thinking through these questions and a precise wording will help future scholars to built upon this work.

This might also help to remove a simple alternative hypothesis that could explain the data equally well: All differences in the comment sections are due to differences in moderation and not the “attraction of toxicity”. In polarized countries, legacy news try to cater to a more split audience and thus moderate less (I am not saying that this is the case, but there is nothing in the theoretical argument or the paper that does say otherwise)

Somewhat related is the (computer science based) use of toxicity. I would encourage the authors to engage with the theoretically more nuanced concept of incivility instead as it seems to be a better fit to their overall framework (Coe et al., 2014; Muddiman & Stroud, 2017; Ziegele et al., 2014)

(2) Methodological adequacy. Overall, the methods seem to be sound. The comparative context is well chosen and justified. The composition of the data sets was really hard to understand, and abbreviations do seldomly make anything better in terms of readability. The study seems to realize a 2 (Canada vs. Brasil) × 2(legacy vs. partisan news) design that is complemented by different robustness checks in the supplementary material. But it took me a separate paper to understand that. This should not be the case. A simple cross table could help to visualize and describe these four main data sets. I also completely lost track when all the other Facebook data sets came into play, and I would encourage the authors to move all (laudable) additional analyses that are not directly relevant to testing the hypotheses in the supplementary material. This would also reduce the readers confusion about all the different platforms used to get the data for the “baseline” comparisons (another term I found highly misleading as the one data set seems to be the toxic extreme rather than the baseline). Furthermore, the actual number of posts in each of the datasets should be clearly visible in that table.

(3) Analytical Rigor. The labeling of the central variable of interest was done algorithmically. As such the authors should manually validate the suitability of the algorithm to detect toxicity and report the results of these robustness checks. This is particularly relevant given the reported (!) bias against non-standard English (p.23) and the intercultural comparison. The preprocessing was done carefully but should be reported in the supplementary section to.

I am less convinced of the chosen statistical approach. First, the authors should provide a table with the means and standard deviations for toxicity in each context. I would also encourage them to think about the smallest effect size of interest (SEIO) for their paper (Lakens, 2017). Their argumentation is very often “yes, there is a difference, but the difference is small” (e.g., p. 27). But small is not necessarily irrelevant, particularly when we talk about news that potentially get millions of views.

After that, they should turn to the main analyses. Here the authors actually had a 2 × 2 design and should test it as such, discussing the main and interaction effects followed by pairwise comparisons (this would also allow them to say: “neutral” vs. “partisan” news have an effect but that is moderated by context, which seems to be what they are interested in). Second, a simple statement that Kruskal-Wallis tests were used to account for the (non-normal) data structure. Then they should report test after test and clearly state if that result was consistent with their hypotheses or not. Currently, the result section is very long and hard to follow.

(4) Minor issues:

• p.1, l. 13 - the authors might want to quote some of the early optimistic sholars believing that the internet would be a "school for deliberative democracy"

• P.2, l. 31 given the numeric referencing system, I would recommend reformulating the sentence and actually name the authors here (as done in the following sentence)

• P. 3, l. 70 surely a question of style but the hypotheses come pretty out of the blue and are not derived from the text that comes before them. That reduces their convincing potential.

• P. 3, l. 78 - I also find it unusal to read the results before the actually paper (but that might be an interdisciplinary issue). Personally, I would delete the abstracts starting l.65 and l.78 and continue directly with the explanation of the papers structure

• Figures should be readably for people who are color blind

• Is it correct that the comments in Brasil legacy news were more toxic than the comparative data set with very toxic Wikipedia comments? (Figure 1)

References

Coe, K., Kenski, K., & Rains, S. A. (2014). Online and uncivil? Patterns and determinants of incivility in newspaper website comments. Journal of Communication, 64(4), 658–679. https://doi.org/10/f6dxrx

Gibson, A. (2019). Free speech and safe spaces: How moderation policies shape online discussion spaces. Social Media + Society, 5(1), 205630511983258. https://doi.org/10.1177/2056305119832588

Habermas, J. (2006). Political communication in media society: Does democracy still enjoy an epistemic dimension? The impact of normative theory on empirical research. Communication Theory, 16, 411–426. https://doi.org/10.1111/j.1468-2885.2006.00280.x

Lakens, D. (2017). Equivalence tests: A practical primer for t tests, correlations, and meta-analyses. Social Psychological and Personality Science, 8(4), 355–362. https://doi.org/10/gbf8nt

Mouffe, C. (2018). For a left populism. Verso Books.

Muddiman, A., & Stroud, N. J. (2017). News values, cognitive biases, and partisan incivility in comment sections. Journal of Communication, 67(4), 586–609. https://doi.org/10/gbr4c8

Ziegele, M., Breiner, T., & Quiring, O. (2014). What creates interactivity in online news discussions? An exploratory analysis of discussion factors in user comments on news items. Journal of Communication, 64(6), 1111–1138. https://doi.org/10.1111/jcom.12123

7. PLOS authors have the option to publish the peer review history of their article (what does this mean?). If published, this will include your full peer review and any attached files.

Reviewer #1: No

Reviewer #3: No

---

## [Author Response · Author response to Decision Letter 1]

29 Dec 2023

Please find attached a document with responses for all comments.

---

## [Decision Letter · Decision Letter 2]

20 Feb 2024

PONE-D-23-09232R2Bubble reachers and uncivil discourse in polarized online public spherePLOS ONE

Dear Dr. Silva,

Thank you for submitting your manuscript to PLOS ONE. After careful consideration, we feel that it has merit but does not fully meet PLOS ONE’s publication criteria as it currently stands. Therefore, we invite you to submit a revised version of the manuscript that addresses the points raised during the review process.

We look forward to receiving your revised manuscript.

Kind regards,

Michal Ptaszynski, PhD

Academic Editor

PLOS ONE

Journal Requirements:

Reviewers' comments:

Reviewer's Responses to Questions

**Comments to the Author**

1. If the authors have adequately addressed your comments raised in a previous round of review and you feel that this manuscript is now acceptable for publication, you may indicate that here to bypass the “Comments to the Author” section, enter your conflict of interest statement in the “Confidential to Editor” section, and submit your "Accept" recommendation.

Reviewer #3: All comments have been addressed

Reviewer #4: (No Response)

2. Is the manuscript technically sound, and do the data support the conclusions?

Reviewer #3: Yes

Reviewer #4: Yes

3. Has the statistical analysis been performed appropriately and rigorously? 

Reviewer #3: Yes

Reviewer #4: Yes

4. Have the authors made all data underlying the findings in their manuscript fully available?

Reviewer #3: Yes

Reviewer #4: No

5. Is the manuscript presented in an intelligible fashion and written in standard English?

Reviewer #3: Yes

Reviewer #4: Yes

6. Review Comments to the Author

Reviewer #3: I want to congratulate the authors to their revision. Although I still think that we don't need more papers that contribute to the already persistent myth of filter bubbles, I acknowledge the authors desire to stick with the metaphor due to the usual language in their discipline. I think that the paper has substantially improved and is now much better grounded in the literature and I think that it will be of great interest to the audience of PLOS One.

Reviewer #4: There is a lot to like about this paper. I would like to particularly highlight the authors‘ profound knowledge of the literature. Different from many other papers on echo chambers and related issues, the authors show a nuanced, differentiated, realistic view on echo-chamber like phenomena and their effects. Altogether, this paper seems to me like the result of a truly successful interdisciplinary collaboration. The methodological design is well-developed and justified and explained in a well-understandable way. I have some comments on how to further improve the paper, but I want to point out that these are not fundamental in nature and can relatively easily be fixed.

The authors seem to have a clear understanding of echo chambers and filter bubbles and the differences between both, which becomes apparent several times in the paper when they implicitly refer to that. However, I think the paper would benefit if the authors clearly defined both terms and explained the difference between these phenomena already in the introduction.

The same applies to populism which is a quite central concept in this study but the paper lacks a definition and conceptualization of it. Please define what you mean with populism and populist communication. And be aware that all political actors make use of populist communication, not only so-called populists (e.g., Ernst et al., 2019; Lilleker & Balaban, 2021). In the same vein, I wonder if H3 and H4 should rather be about contexts that are MORE populist and polarized. As they read now, it sounds like there was a binary distinction between populist/polarized on the one hand and non-populist/non-polarized contexts on the other hand. However, both populism and polarization are rather continuums on which countries can be more or less populist/polarized. Regarding the example of populism, it is worth mentioning that anti-elitism and being rather exclusive is typical for populism in Latin America, different from the US and Europe (Rovira Kaltwasser, 2012). It would also be a good idea, I think, to mention that anti-elitism is often typical for left-wing populism while right-wing populism is often rather characterized by excluding outgroups (Reinemann et al., 2017).

Rightly, the authors point out that bubbles or echo chambers let in contrary content but it is inside the bubble not presented in a neutral way but rather “filtered through a partisan lense“ (lines 62-63). I would like to recommend a paper by Bright et al. (2020) to the authors which may help to further strengthen this argument.

The term “platform affordances“ is a quite vague one with many different meanings and understandings. Therefore, I recommend avoiding this term and replacing it by something different (e.g., platform characteristics).

Unfortunately, I don’t understand the following sentence: “As Alexander [47] observes, populists portray claims to impartial expertise by mediating institutions as a cover for narrower interests of the status quo.“ Please make clearer what you mean with this.

I agree that Brazil and Canada are good cases for the comparison in this paper. However, the country selection would be even more convincing if the authors could give some more information about structural similarities/differences between both countries: Is this a most similar systems design with the different degrees of populism and polarization being the only probable reasons for the differences found? Or might there be alternative structural explanations? This should also include some numbers on the use of Facebook and other relevant media.

Regarding the literature review, the authors show in-depth knowledge on computational studies in their field, but I think it would be good to add some manual content analyses on incivility in user comments.

The dataset PARTISAN_pt, as I understand, only includes conservative Facebook pages. Is it a problem that it does not contain any left-wing pages? Regarding CIVIL_pt and CIVIL_en, I wonder why the authors did not use equivalent websites in both countries and how this might affect the cross-country comparability of their data.

For the examples given in the text which are in Portuguese, the authors should give English translations so that all readers can understand them.

When it comes to the period under investigation, I wonder if it might have influenced the findings that the data were collected in the midth of the Covid-19 pandemic which polarized the public discourse in many countries. This is of course a hypothetical question but I think it is worth addressing it, for example in the discussion: Is it conceivable that the findings would have been different in a more routine phase?

When it comes to news media as bubble reachers, the authors‘ argument could be strengthened by a study by Magin et al. (2021) which shows that news media helps people with more extreme opinions to maintain the connection with the common core of the public discourse.

Regarding the findings, I wonder if differences between different news media types might help explaining them, e.g. the difference between quality newspapers and tabloids or the difference between legacy media and alternative media. Maybe the authors could discuss that in the conclusion.

The explanation why the authors use one dataset including US media is first given in the discussion. This should be moved to the methods section.

Line 764: “uncivil“ must be “uncivility“.

References

Bright, J., N. Marchal, B. Ganesh, and S. Rudinac. 2020. Echo Chambers Exist! (But They’re Full of Opposing Views). doi: https://arxiv.org/abs/2001.11461

Ernst, N., Esser, F., Blassnig, S., & Engesser, S. (2019). Favorable opportunity structures for populist communication: Comparing different types of politicians and issues in social media, television and the press. The International Journal of Press/Politics, 24(2), 165–188. https://doi.org/10.1177/1940161218819430

Lilleker, D. G., & Balaban, D. (2021). Populism on Facebook. In J. Haßler, M. Magin, U. Rußmann, & V. Fenoll (Eds.), Campaigning on Facebook in the 2019 European parliament election: Informing, interacting with, and mobilising voters (pp. 267–282). Palgrave Macmillan. https://doi.org/10.1007/978-3-030-73851-8_17

Magin, M., Geiß, S., Stark, B. & Jürgens, P. (2022). Common Core in Danger? Personalized Information and the Fragmentation of the Public Agenda. The International Journal of Press/Politics 17(4), 887-909. https://doi.org/10.1177/19401612211026595

Reinemann, C., Aalberg, T., Esser, F., Strömbäck, J., & de Vreese, C. (2017). Populist political communication: Toward a model of its causes, forms, and effects. In T. Aalberg, F. Esser, C. Reinemann, J. Strömbäck, & C. de Vreese (Eds.), Populist political communication in Europe (pp. 12–25). Routledge. https://doi.org/10.4324/9781315623016

Rovira Kaltwasser, C. R. (2012). The ambivalence of populism: Threat and corrective for democracy. Democratization, 19(2), 184–208. https://doi.org/10.1080/13510347.2011.572619

7. PLOS authors have the option to publish the peer review history of their article (what does this mean?). If published, this will include your full peer review and any attached files.

Reviewer #3: No

Reviewer #4: No

---

## [Author Response · Author response to Decision Letter 2]

25 Mar 2024

Dear reviewers,

We would like to thank you for reviewing our manuscript. We have revised the paper taking into account the Reviewers’ remarks and suggestions, which were quite helpful. Please, refer to the "Response to Reviewers" PDF for the complete response.

---

## [Decision Letter · Decision Letter 3]

9 May 2024

PONE-D-23-09232R3Bubble reachers and uncivil discourse in polarized online public spherePLOS ONE

Dear Dr. Silva,

Thank you for submitting your manuscript to PLOS ONE. After careful consideration, we feel that it has merit but does not fully meet PLOS ONE’s publication criteria as it currently stands. Therefore, we invite you to submit a revised version of the manuscript that addresses the points raised during the review process.

The Academic Editor is satisfied, based on their own assessment and the reviewer comments below, that you have addressed all scientific concerns and your submission is ready for publication. Before we can proceed, we noted during editorial checks that there remain some instances of (censored) offensive language in your manuscript text, specifically on lines 770-777 of the current 'clean' manuscript file. We apologize that these were not identified and brought to your attention to address sooner. While the Academic Editor does not object to the use of these quoted examples, in order to meet the journal editorial requirements, as previously discussed, we ask you to remove these quoted examples from your manuscript text. We otherwise appreciate your efforts to make the full uncensored dataset separately available.  Following discussion with the Academic Editor, we do not anticipate any further peer review to be necessary; aside from this minor edit to the manuscript text, there are no further matters to address before proceeding with publication.

We look forward to receiving your revised manuscript.

Kind regards,

Hugh Cowley

Senior Editor

PLOS ONE

on behalf of

Michal Ptaszynski, PhD

Academic Editor

PLOS ONE

Journal Requirements:

Reviewers' comments:

Reviewer's Responses to Questions

**Comments to the Author**

1. If the authors have adequately addressed your comments raised in a previous round of review and you feel that this manuscript is now acceptable for publication, you may indicate that here to bypass the “Comments to the Author” section, enter your conflict of interest statement in the “Confidential to Editor” section, and submit your "Accept" recommendation.

Reviewer #1: All comments have been addressed

Reviewer #4: All comments have been addressed

2. Is the manuscript technically sound, and do the data support the conclusions?

Reviewer #1: Yes

Reviewer #4: Yes

3. Has the statistical analysis been performed appropriately and rigorously? 

Reviewer #1: Yes

Reviewer #4: I Don't Know

4. Have the authors made all data underlying the findings in their manuscript fully available?

Reviewer #1: Yes

Reviewer #4: Yes

5. Is the manuscript presented in an intelligible fashion and written in standard English?

Reviewer #1: Yes

Reviewer #4: Yes

6. Review Comments to the Author

Reviewer #1: I have re-reviewed the manuscript twice now, and I feel that the authors have addressed all of my concerns (and those of other reviewers) excellently. The paper makes a strong contribution and I'm happy to recommend it for publication.

Reviewer #4: i would like to thank the authors for considering and implementing my comments in a convincing manner. I think the manuscript has further improved in quality and is now ready for publication.

7. PLOS authors have the option to publish the peer review history of their article (what does this mean?). If published, this will include your full peer review and any attached files.

Reviewer #1: No

Reviewer #4: No

---

## [Author Response · Author response to Decision Letter 3]

10 May 2024

Dear reviewers, 

we greatly appreciate the time and effort you have dedicated to this review process, we believe the article has undergone significant improvement and is now well-prepared for publication.

---

## [Editor Report · Decision Letter 4]

15 May 2024

Bubble reachers and uncivil discourse in polarized online public sphere

PONE-D-23-09232R4

Dear Dr. Silva,

We’re pleased to inform you that your manuscript has been judged scientifically suitable for publication and will be formally accepted for publication once it meets all outstanding technical requirements.

Kind regards,

Michal Ptaszynski, PhD

Academic Editor

PLOS ONE
---

## [Editor Report · Acceptance letter]

24 May 2024

PONE-D-23-09232R4 

PLOS ONE

Dear Dr. Silva, 

I'm pleased to inform you that your manuscript has been deemed suitable for publication in PLOS ONE. Congratulations! Your manuscript is now being handed over to our production team.

Kind regards, 

on behalf of

Dr. Michal Ptaszynski 

Academic Editor

PLOS ONE